**Investigation**

# A single mutation G454A in the P450 *CYP9K1* drives pyrethroid resistance in the major malaria vector *Anopheles funestus* reducing bed net efficacy

Carlos S. Djoko Tagne [iD] ,[1,2,*] Mersimine F.M. Kouamo,[1] Magellan Tchouakui,[1] Abdullahi Muhammad,[3,4]
Leon J.L. Mugenzi,[5] Nelly M.T. Tatchou-Nebangwa,[1,6] Riccado F. Thiomela,[1] Mahamat Gadji [iD] ,[1] Murielle J. Wondji,[1,3]
Jack Hearn,[7] Mbouobda H. Desire,[2] Sulaiman S. Ibrahim [iD] ,[1,8] Charles S. Wondji [iD] [1,3,*]

[1]Medical Entomology Department, Centre for Research in Infectious Diseases (CRID), P.O. Box 13501, Yaoundé, Cameroon
[2]Department of Biochemistry, Faculty of Science, University of Bamenda, P.O. Box 39 Bambili, Bamenda, Cameroon
[3]Vector Biology Department, Liverpool School of Tropical Medicine, Pembroke Place, Liverpool L3 5QA, UK
[4]Centre for Biotechnology Research, Bayero University, Kano, PMB 3011, Kano, Nigeria
[5]Syngenta Crop Protection Department, Werk Stein, Schaffhauserstrasse, Stein CH4332, Switzerland
[6]Department of Biochemistry and Molecular Biology, Faculty of Science, University of Buea, P.O Box 63, Buea, Cameroon
[7]Centre for Epidemiology and Planetary Health, Scotland's Rural College (SRUC), RAVIC, Inverness IV2 5NA, UK
[8]Department of Biochemistry, Bayero University, PMB 3011 Kano, Nigeria

*Corresponding authors: Charles S. Wondji, Liverpool School of Tropical Medicine (LSTM), Pembroke Place, Liverpool L3 5QA, UK. Email: charles.wondji@lstmed.ac.uk;
Carlos S. Djoko Tagne, Medical Entomology Department, Centre for Research in Infectious Diseases (CRID), P.O. Box 13501, Yaounde, Cameroon; Department of
Biochemistry, Faculty of Science, University of Bamenda, P.O Box 39 Bambili, Bamenda, Cameroon. Email: carlos.djoko@crid-cam.net

Metabolic mechanisms conferring pyrethroid resistance in malaria vectors are jeopardizing the effectiveness of insecticide-based interventions, and identification of their markers is a key requirement for robust resistance management. Here, using a field-lab-field approach, we demonstrated that a single mutation G454A in the P450 *CYP9K1* is driving pyrethroid resistance in the major malaria vector *Anopheles funestus* in East and Central Africa. Drastic reduction in *CYP9K1* diversity was observed in Ugandan samples collected in 2014, with the selection of a predominant haplotype (G454A mutation at 90%), which was completely absent in the other African regions. However, 6 years later (2020) the Ugandan 454A-*CYP9K1* haplotype was found predominant in Cameroon (84.6%), but absent in Malawi (Southern Africa) and Ghana (West Africa). Comparative *in vitro* heterologous expression and metabolism assays revealed that the mutant 454A-*CYP9K1* (R) allele significantly metabolizes more type II pyrethroid (deltamethrin) compared with the wild G454-*CYP9K1* (S) allele. Transgenic *Drosophila melanogaster* flies expressing 454A-*CYP9K1* (R) allele exhibited significantly higher type I and II pyrethroids resistance compared to flies expressing the wild G454-*CYP9K1* (S) allele. Furthermore, laboratory testing and field experimental hut trials in Cameroon demonstrated that mosquitoes harboring the resistant 454A-*CYP9K1* allele significantly survived pyrethroids exposure (odds ratio = 567, $P < 0.0001$). This study highlights the rapid spread of pyrethroid-resistant *CYP9K1* allele, under directional selection in East and Central Africa, contributing to reduced bed net efficacy. The newly designed DNA-based assay here will add to the toolbox of resistance monitoring and improving its management strategies.

## Graphical abstract

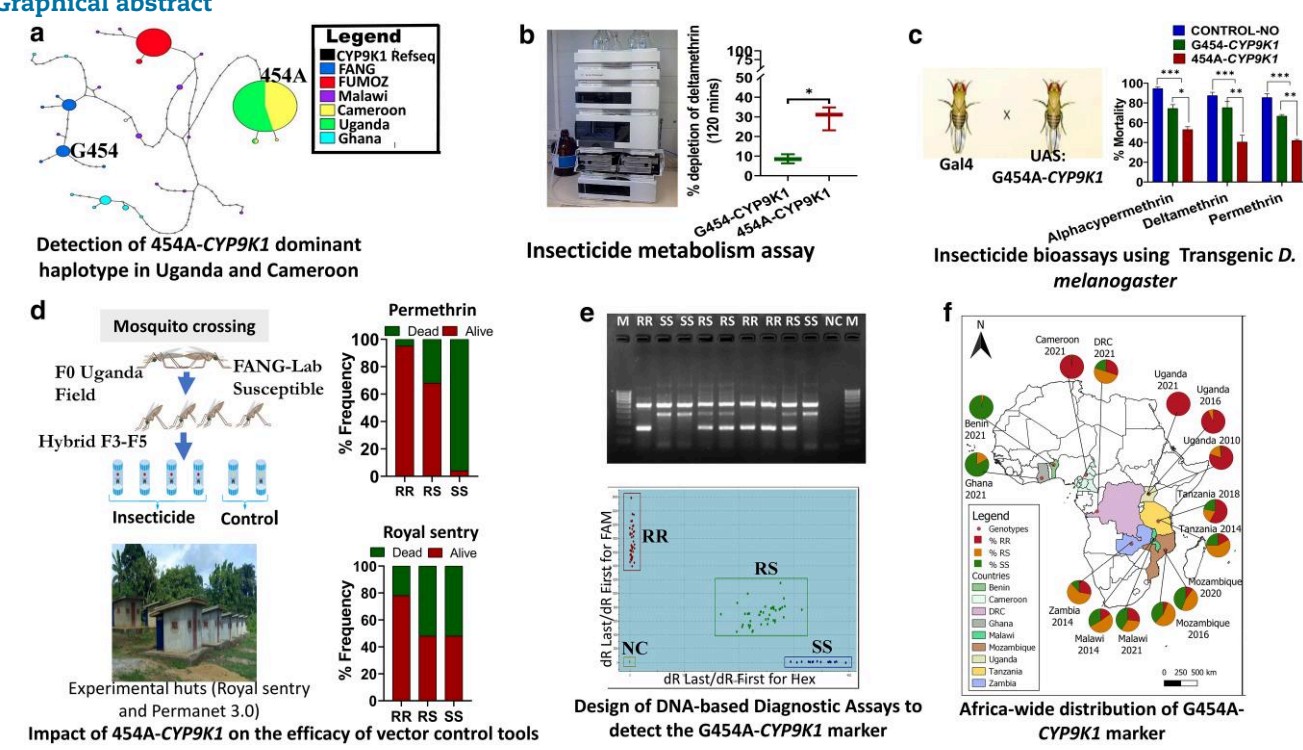

Keywords: *CYP9K1*; pyrethroid; insecticide resistance; insecticide-treated nets; *Anopheles funestus*; malaria; bed net efficacy

## Introduction

Malaria is still a major public health concern despite efforts to reduce its burden (WHO 2023). Globally, an increase in the number of cases was recently recorded from 233 to 249 million cases respectively in 2019 and 2022, with 95% of cases in Africa (WHO 2023). Vector control, mainly using long-lasting insecticidal nets (LLINs) and indoor residual spraying (IRS) is a frontline approach in use to fight malaria. During the past 2 decades, nearly 2.5 billion LLINs were delivered to malaria-endemic countries and this rapid scale-up has been by far the largest contributor to the drops seen in malaria incidence and mortality since the turn of the century (WHO 2022). These control tools contributed more than 70% to the decrease in malaria mortality and helped prevent more than 663 million clinical cases between 2000 and 2015 (Bhatt *et al.* 2015).

Unfortunately, sub-Saharan African countries bearing the highest malaria burden are failing to maintain the trajectory of the WHO global technical strategy (GTS) milestones, which aims to reduce malaria morbidity and mortality to 244 cases and 6.7 deaths per 100,000 population at risk by 2030 (WHO 2023). Several factors contribute to this lack of progress in sub-Saharan Africa, including the continuous spread and escalation of resistance to pyrethroid insecticide in major malaria vectors (Ibrahim *et al.* 2019; Riveron *et al.* 2019; Muhammad *et al.* 2021; Tchouakui *et al.* 2021; Menze, Tchouakui, *et al.* 2022; Mugenzi *et al.* 2022; Tepa *et al.* 2022; Nguiffo-Nguete *et al.* 2023). This phenomenon presents a greater risk of vector control failure (Hemingway 2017). Moreover, recent studies have highlighted that pyrethroid-resistant mosquitoes present a serious challenge to the efficacy of current standard pyrethroid LLINs (Mugenzi *et al.* 2019; Riveron *et al.* 2019; Weedall *et al.*

2019; Menze *et al.* 2020; Menze, Mugenzi, *et al.* 2022). Pyrethroid resistance is conferred by several mechanisms, but primarily by increased expression and overactivity of metabolic detoxification resistance genes, including the cytochrome P450s, glutathione-S transferases, and carboxylesterases (Ranson *et al.* 2011; Coetzee and Koekemoer 2013; Weedall *et al.* 2019; Kouamo *et al.* 2021). Additionally, modification of insecticide targets by mutations has been shown to contribute to resistance in Anopheles mosquitoes. For example, the knockdown resistance (*kdr*) in the voltage-gated sodium channels reduces the efficacy of the nerve agents, including pyrethroids and dichlorodiphenyltrichloroethane (DDT) in the major malaria vector *An. gambiae/An. coluzzii* (Martinez-Torres *et al.* 1998; Ranson *et al.* 2000), and modification of the acetylcholine esterase gene (*ace-1* mutation) has been shown in several species to confer insensitivity to organophosphate and carbamate insecticides (Weill *et al.* 2004; Ibrahim, Ndula, *et al.* 2016; Ibrahim, Riveron, *et al.* 2016; Elanga-Ndille *et al.* 2019). Other mechanisms include behavioral changes to avoid insecticide contact (Kreppel *et al.* 2020) and reduced insecticide penetration through increased production of cuticular hydrocarbon (Balabanidou *et al.* 2016). Knockdown resistance, which is highly prevalent and drives resistance in *An. gambiae*, has been absent in *An. funestus* (Hemingway 2014; Irving and Wondji 2017) until the recent report of a L976F-*kdr* in Tanzania. This mutation is associated with resistance to DDT, not pyrethroids (Odero *et al.* 2024), indicating a more pronounced role of metabolic resistance toward pyrethroids in this species.

Previous transcriptional profiling studies have identified key *An. funestus* cytochrome P450 genes with differential expressions linked to pyrethroid resistance (Riveron *et al.* 2017;

Mugenzi *et al.* 2019; Weedall *et al.* 2019). However, significant regional contrasts were reported with different *CYP* genes overexpressed in various African regions (Riveron *et al.* 2017; Weedall *et al.* 2019). Among these genes, the duplicated *CYP6P9a/b* and *CYP6P4a/b* were overexpressed in *An. funestus* mosquitoes from Southern (Malawi and Mozambique) and West (Ghana) African regions, respectively. In Central Africa (Cameroon), *CYP325A* was overexpressed, while *CYP9K1* was the most up regulated gene in resistant mosquitoes from East Africa (Uganda) (Mulamba *et al.* 2014; Riveron *et al.* 2017; Weedall *et al.* 2019). The molecular bases of resistance mediated by these P450s are gradually being deciphered. For instance, *CYP325A* contributes to resistance to both type I and II pyrethroid insecticides in central Africa (Wamba *et al.* 2021). Functional characterization has also shown that allelic variants of *CYP6P9a* and *CYP6P9b* are major drivers of type I and type II pyrethroid resistance, primarily in Southern Africa (Riveron *et al.* 2013; Ibrahim *et al.* 2015). Further studies targeting the promoter and intergenic regions of the key P450s *CYP6P9a* and *CYP6P9b* have confirmed that these duplicated genes are the main drivers of pyrethroid resistance in *An. funestus*, (Mugenzi *et al.* 2019, 2020; Weedall *et al.* 2019), leading to the detection of the first P450-based molecular markers. However, the resistance driven by the *CYP6P9a/b* genes is limited to the southern African region, and the assays designed on the detected markers of these genes are mainly used to track resistance in this region. Therefore, there is an urgent need to identify the markers driving resistance in other African regions to facilitate resistance monitoring and management in these regions.

An Africa-wide whole genome scan and targeted enrichment with deep sequencing of permethrin-resistant *An. funestus* mosquitoes have previously reported reduced genetic diversity with the signature of directional selection and gene duplication on the X-chromosome spanning the *CYP9K1* locus (Weedall *et al.* 2020; Hearn *et al.* 2022). Further analysis of the genetic diversity around the *CYP9K1* gene across Africa identified a glycine to alanine amino acid change in codon 454, that was fixed in Uganda, while present at very low frequencies in other African regions (Cameroon and Malawi) (Hearn *et al.* 2022). Here, we used a field-laboratory-field approach to unveil the role of the P450 *CYP9K1* in pyrethroid resistance in *An. funestus*. After establishing the genetic polymorphism of the *CYP9K1* gene in field-resistant *An. funestus* mosquitoes across Africa, we next used comparative in vitro heterologous metabolism assays and in vivo transgenic *Drosophila* fly approaches to assess the contribution of allelic variation and overexpression of this gene to the observed resistance. We then designed a simple DNA-based diagnostic assay targeting the G454A mutation and assessed the impact of this mutation on the efficacy of vector control tools using LLINs cone assays in the lab and field experimental hut trials (EHTs) on free-flying mosquito populations. We established the distribution and patterns of evolution of this G454A-*CYP9K1* marker in *An. funestus* mosquitoes from different African regions, contributing to the surveillance of metabolic resistance in these regions. This study offers a new DNA-based molecular diagnostic tool for the real-time monitoring of metabolic resistance in *An. funestus* mosquitoes, improving vector control strategies through evidence-based and timely decision-making.

## Materials and methods

Key resources used in this research are listed in Supplementary Table 1.

## Investigation of genetic variability of *An. funestus* CYP9K1 gene across Africa and evaluation of temporal changes in allele frequencies

### Mosquito sampling and rearing

The mosquitoes used in this study were collected from 4 different geographical regions of sub-Saharan Africa (see Fig. 5a). Field *An. funestus* mosquitoes were collected from Uganda (East Africa), in Tororo district (0°45′N, 34°5′E) in March 2014 (Mulamba *et al.* 2014) and in Mayuge (0°23010.8′N, 33°37016.5′E) in September 2020 (Tchouakui *et al.* 2021); from Malawi (Southern Africa), in Chikwawa (12°19′S, 34°01′E) in January 2014 (Riveron *et al.* 2015) and in June 2021 (Menze, Tchouakui, *et al.* 2022); from Cameroon (Central Africa), in Mibellon (6°46′N, 11°70′E) in February 2015 (Menze *et al.* 2018) and in October 2020, and Elende (3°41′ 57.27′N, 11°33′28.46′E) in April 2019 (Nkemngo *et al.* 2020); and from Ghana (West Africa), in Atatam village close to Obuasi (06° 17.377′ N, 001°27.545′ W) in July and in October 2021 (Mugenzi *et al.* 2022) (Fig. 5a). The 2 *An. funestus* laboratory colonies used in this study were the FANG strain (which is fully susceptible to all insecticide classes) from the Calueque district of Southern Angola (16°45′S, 15°7′E) (Hunt *et al.* 2005) and FUMOZ strain from southern Mozambique, which is highly resistant to pyrethroids and carbamates (Hunt *et al.* 2005). Adult blood-fed female *An. funestus* mosquitoes were collected and reared as previously described (Morgan *et al.* 2010). Details of the species identification of these *An. funestus* s.s populations and pyrethroid resistance profiles have been established in previous studies. (Menze *et al.* 2018; Riveron *et al.* 2019; Nkemngo *et al.* 2020; Tchouakui *et al.* 2021; Menze, Tchouakui, *et al.* 2022; Mugenzi *et al.* 2022).

### Africa-wide temporal polymorphism analysis of CYP9K1

To assess the association between genetic diversity and resistance to insecticide, the temporal polymorphism pattern of the *CYP9K1* gene was analyzed across Africa. First, using target enrichment with deep sequencing (Sure Select) data for permethrin-resistant samples collected in 2014 across Africa was previously described (Hearn *et al.* 2022). Secondly, the full-length cDNA of this gene was amplified from permethrin-resistant *An. funestus* mosquitoes collected in 2020 from Uganda (Mayuge), Cameroon (Mibellon), Ghana (Obuasi), and Malawi (Chikwawa). To provide additional contrast in the sequence analysis of the *CYP9K1* gene in different *An. funestus* populations, we used 2 laboratory-maintained *An. funestus* mosquito strains FANG and FUMOZ, respectively, susceptible and resistant to pyrethroid as described in previous studies (Riveron *et al.* 2013; Ibrahim *et al.* 2015; Wamba *et al.* 2021). Detailed procedures are provided in the Supplementary material, and a list of primers are provided in Supplementary Table 2.

## Assessment of the association between the G454A-CYP9K1 mutation and insecticide resistance through *in vitro* and *in vivo* experiments

### Heterologous expression of recombinant CYP9K1 allelic variants and metabolism assays

Recombinant enzymes of the mutant-type 454A-*CYP9K1* and wild-type G454-*CYP9K1* alleles were expressed as previously described for other P450s with slight modifications (Riveron *et al.* 2013; Ibrahim *et al.* 2015; Wamba *et al.* 2021). The modifications were mainly made on the primers designed and restriction enzyme sites used for the cloning of the candidate alleles. Expression plasmids *pB13::ompA + 2-CYP9K1* for both alleles of

mutant-type 454A-*CYP9K1* allele from Uganda and Cameroon and the wild-type G454-*CYP9K1* allele from FANG were constructed by fusing cDNA fragment from a bacterial ompA + 2 leader sequence with its downstream ala-pro linker to the $NH_2$-terminus of the P450 cDNA, in frame with the P450 initiation codon, as previously described (Pritchard *et al.* 1997). This was then cloned into *NdeI* and *EcoRI* linearized *pCW-ori* + expression vector (McLaughlin *et al.* 2008). Each *CYP9K1* allele was co-transformed into *E. coli* together with the recently expressed *An. funestus* cytochrome P450 reductase (Ibrahim *et al.* 2024; Tchouakui *et al.* 2024). Membrane expression, preparations, and measurement of P450 content were carried out as previously described (Stevenson *et al.* 2011; Riveron *et al.* 2013; Ibrahim *et al.* 2015) with slight modifications. Primers used for in vitro expression are listed in Supplementary Table 2.

Metabolism assays were conducted using the recombinant enzymes of *CYP9K1* alleles with permethrin (type I pyrethroid) and deltamethrin (type II pyrethroid) following protocols described previously with some modifications (Riveron *et al.* 2013; Ibrahim *et al.* 2015; Hearn *et al.* 2022). Briefly, 0.2 M of potassium phosphate buffer (pH 7.4) and NADPH-regeneration components (1 mM glucose-6-phosphate, 0.25 mM $MgCl_2$, and 0.1 mM NADP and 1 U/ml glucose-6-phosphate dehydrogenase) were added to the bottom of 1.5-ml tube chilled on ice. Membrane expressing recombinant G454-*CYP9K1* (wild allele) and 454A-*CYP9K1* (mutant allele), and cytochrome $b_5$ were added to the side of the tube. These were incubated for 5 min at 30 °C, with shaking at 1,200 rpm to activate the membrane; 20 μM of insecticides were added into the final volume of 0.2 ml (~2.5% v/v acetonitrile), and the reaction started by vortexing at 1,200 rpm and 30 °C for 1 h. Reactions were quenched with 0.1 ml ice-cold acetonitrile and incubated for 5 more mins. Tubes were then centrifuged at 16,000 rpm and 4 °C for 15 min, and 150 μl of supernatant was transferred into HPLC vials for analysis. Reactions were carried out in triplicates with experimental samples (+NADPH) and negative controls (−NADPH); 100 μl of sample was loaded onto an isocratic mobile phase (90:10 v/v acetonitrile to water) with a flow rate of 1 ml/min, monitoring wavelength of 226 nm and peaks separated with a 250-mm C18 column (Acclaim 120, Dionex) on Agilent 1260 Infinity at 23 °C. Enzyme activity was calculated as percentage depletion (the difference in the amount of insecticide remaining in the +NADPH tubes compared with the −NADPH).

### Comparative in vivo assessment of the ability of CYP9K1 allelic variants to confer pyrethroid resistance using transgenic Drosophila flies

To investigate if overexpressing the *CYP9K1* gene alone could drive resistance to pyrethroid, and to determine the phenotypic impact of the G454A mutation, 2 *CYP9K1 D. melanogaster* transgenic lines were constructed: 454A-*CYP9K1-UAS-UGA* expressing the mutant field allele and G454-*CYP9K1-UAS-FANG* expressing the wild allele. Cloning techniques, construction of transgenic flies, and contact bioassays with transgenic flies have been described in previous studies (Riveron *et al.* 2013; Riveron, Cristina, *et al.* 2014; Riveron, Ibrahim, *et al.* 2014; Ibrahim *et al.* 2015). To create the transgenic flies, the *Drosophila melanogaster* ("y1w67c23; P attP40 25C6," "1;2") lines obtained from the University of Cambridge, UK, were injected with *CYP9K1* alleles. Virgin females from these lines were crossed with males from the Act5C-GAL4 strain ("y[1] w[*]; P(Act5CGAL4-w)E1/CyO," "1;2") obtained from the Bloomington Stock Centre, USA. Detailed procedures for cloning *CYP9K1* alleles and maintaining transgenic fly groups are elaborated in the

Supplementary material, and the primers utilized are listed in Supplementary Table 2.

The $F_1$ progenies expressing the above mutant and wild alleles were used for contact bioassays. Insecticides (0.007% alphacypermethrin, 0.2% deltamethrin and 4% permethrin) were impregnated onto filter papers prepared in acetone and Dow Corning 556 Silicone Fluid (BDH/Merk, Germany) and stored at 4 °C prior to bioassay (Riveron *et al.* 2013). These papers were rolled and introduced into 45 cc plastic vials, which were then plugged with cotton wool soaked in 5% sucrose. Six replicates of 20–25 female *D. melanogaster* flies, 2–4-d-old post-eclosion, were introduced into the vials. Mortality plus knockdown were recorded after 1, 2, 3, 6, 12, and 24 h of exposure to the above insecticides.

To confirm the expression of the *CYP9K1* alleles in the 2 experimental fly groups compared to the control group (flies not expressing the P450), qRT-PCR was performed as described previously (Riveron *et al.* 2013; Riveron, Cristina, *et al.* 2014; Riveron, Ibrahim, *et al.* 2014). RNA was extracted separately from 3 pools of 5 $F_1$ flies of experimental and control flies and used for cDNA synthesis. The qRT-PCR was conducted using the primers listed in Supplementary Table 2, with normalization using the housekeeping gene *RPL11*.

## Design of a DNA-based diagnostic assay for the G454A-CYP9K1 mutation and assessment of its correlation with pyrethroid resistance

To facilitate the detection and monitoring of resistance driven by the *CYP9K1* gene, 2 DNA-based PCR diagnostic assays: an allele-specific PCR (AS-PCR) assay and a probe-based locked nucleic acid (LNA) assay were designed for the G/C-*CYP9K1* nucleotide polymorphism at position 1361 in the coding sequence of this gene.

### AS-PCR design

The *CYP9K1* AS-PCR assay was designed to target the G454A mutation as described in a previous study for another marker (Tchouakui *et al.* 2019). The AS-PCR utilizes 2 pairs of manually designed primers: a pair of outer primers comprising of an outer forward primer (9K1_OF: 5′-ACTGGACCGATGATGATTTGAC-3′) and an outer reverse primer (9K1_OR: 5′-ATCCAGAAGCCCTTCTCTGC-3′) and a pair of inner primers designed to match the mutation. An additional mismatched nucleotide (underlined) was added in the third nucleotide from the 3′ end of each inner primer (9K1_IF: 5′-GGATCGTTTCTGGCCGGAAGGTTGG**C-3**′ and 9K1_IR: 5′-TATCGATCGGTGTCGGGCTGTCCGC**T C-3**′) to enhance the specificity (Supplementary Table 3). Detailed AS-PCR reagents and conditions are elaborated on in the Supplemental material.

### Locked nucleic acid assay design

An alternative LNA assay was also designed targeting the G454A-*CYP9K1* mutation with LNA probes for potentially improved performance. This assay design followed previously established protocols (Mouritzen *et al.* 2003). LNA primers and probes (provided in Supplementary Table 3) were synthesized by integrated DNA technologies (IDT) (www.idtdna.com). The *CYP9K1*-LNA assay employs LNA probes (LNA9K1-Gly: Hex: TCCGG + T + C + CGAAC and LNA9K1-Ala: Fam: TCCGG + T + G + CG + AA) and primers (LNA-9K1F: 5′-CGTGATCCGCAACTGTTTC-3′ and LNA-9K1R: 5′-GTAAGGAT GGACGCGGTATC-3′). Detailed LNA PCR reagents and conditions are provided in the Supplemental material.

## Assessment of correlation between the G454A-CYP9K1 mutation and resistance phenotype

*Correlation using hybrid strains.* Two genetic mosquito crosses were created: (i) between the susceptible female FANG and the field male resistant mosquitoes from Mibellon maintained to the third generation (FANG × Mibellon $F_3$), and (ii) between FANG females and field from Mayuge, which were maintained to the fifth generation (FANG × Mayuge $F_5$). Crossing protocols were as previously described (Wondji *et al.* 2007). WHO tube bioassays (WHO 2016) were conducted on 2–5-d-old female progenies from FANG × Mibellon $F_3$ and FANG × Mayuge $F_5$, respectively, using 0.05% alphacypermethrin and 0.75% permethrin. For each of these insecticides, 4 replicates of 25 female mosquitoes were exposed. Mosquitoes were sorted into contrasting phenotypes by separating those that were resistant (alive at 24 h after 1 h of exposure), from mosquitoes that were highly susceptible (dead at 24 h after 10 and 30 min of exposure). Genomic DNA was extracted from these hybrid samples (Livak 1984), and the gDNA samples were genotyped to establish the correlation between the G454A genotype and resistance.

*Correlation using field mosquito strains.* WHO tube bioassays were conducted using mosquitoes which were collected from Elende in 2022. As in the above section 4 replicates of 20–25 $F_1$ female mosquitoes, aged 2–5 d, were exposed to 0.05% alphacypermethrin, 0.05% deltamethrin, and 0.75% permethrin according to the WHO guidelines (WHO 2016). The mosquitoes were exposed for 1 h and transferred into holding tubes to record mortalities after 24 h. Genomic DNA was extracted from both alive and dead mosquitoes and used to assess the correlation between the G454A-*CYP9K1* mutation and pyrethroid resistance.

## Assessment of the impact of G454A-CYP9K1 mutation on the efficacy of LLINs

The efficacy of several standard pyrethroid and combination piperonyl butoxide (PBO)-pyrethroid nets was evaluated using $F_1$ mosquitoes collected from Elende. Cone bioassays were carried out as advised by the WHO (WHO 2013). The nets tested included PermaNet 2.0 (containing deltamethrin), Duranet (containing alphacypermethrin), Royal sentry (containing alphacypermethrin), Olyset (containing permethrin), Olyset plus (containing permethrin and PBO), PermaNet 3.0 top panels (containing deltamethrin and PBO), and PermaNet 3.0 side panels (containing only deltamethrin). Five replicates of 10 mosquitoes were exposed to 30 cm × 30 cm pieces of each net through the cone for 3 mins and thereafter transferred into paper cups with 10% sugar. Knockdown was recorded 1 h after exposure and the mortality was recorded at 24 h. Controls consisted of 4 replicates of 10 mosquitoes exposed to 2 pieces of untreated nets.

## Evaluation of the impact of 454A-CYP9K1 mutation on the efficacy of insecticide-treated bed nets in the field and the spatiotemporal distribution of the marker

*Assessment of the impact of G454A-CYP9K1 marker on LLINs through hut trials*

The experimental hut study was conducted in Elende where 12 West African-type experimental huts (WHO 2013) were recently constructed with concrete bricks and a corrugated aluminum roof. Three treatments were compared in this study: (i) an untreated polyethylene net; (ii) a standard net, Royal sentry (alphacypermethrin impregnated polyethylene net); and (iii) a PBO-net, PermaNet 3.0 (PBO and deltamethrin impregnated into polyethylene net). To simulate a worn net, six 4 cm × 4 cm holes were made on each net according to WHO guidelines. Three adult volunteers from Elende village were recruited to sleep under the nets and collect mosquitoes in the morning. Each volunteer provided written consent and received chemoprophylaxis during the trial. Ethical approval for this study was obtained from the national ethics committee for health research of Cameroon (ID: 2021/07/1372/CE/CNERSH/SP). Early in the morning, mosquitoes were collected using glass tubes from the room (including the floors, walls, and roof of the hut), inside the bed net, and from the exit traps on the veranda. Surviving mosquitoes were provided with sugar solution and held for 24 h in paper cups after which delayed mortality was assessed. Samples were recorded in observation sheets as dead/blood-fed, alive/blood-fed, dead/unfed, and alive/unfed. The effect of each treatment was expressed relative to the control by assessing the killing effect.

## Genotyping of G454A-CYP9K1 mutation in mosquitoes collected from huts

To assess the impact of the 454A-*CYP9K1* mutation on the efficacy of the LLINs, mosquitoes which exhibited contrasting phenotypes from exposure to Royal sentry and PermaNet 3.0 were genotyped using the AS-PCR assay described above. The mosquitoes were the dead and alive mosquitoes collected from the veranda, within the nets, and inside the huts.

## Africa-wide spatiotemporal assessment of the spread of G454A-CYP9K1 marker

To assess the spread and temporal changes in the frequency of the G454A-*CYP9K1* marker in Africa, *An. funestus* samples previously collected at different time points in the same localities across Africa (Menze *et al.* 2018; Nkemngo *et al.* 2020; Weedall *et al.* 2020; Tchouakui *et al.* 2021; Menze, Tchouakui, *et al.* 2022; Mugenzi *et al.* 2022; Nguiffo-Nguete *et al.* 2023) were screened. These were samples from East Africa (Uganda in 2010, 2016, and 2021 and Tanzania in 2014 and 2018), Central Africa (Cameroon (Mibellon in 2016, 2018, and 2021, Elende in 2019, 2021, and 2023, Gounougou in 2017 and 2021, Tibati in 2021, Penja in 2021) and DRC in 2021), Southern Africa (Zambia in 2014, Malawi in 2014, and 2021 and Mozambique in 2016 and 2020) and West Africa (Benin in 2021 and Ghana in 20121). Genomic DNA from these samples was genotyped using the above AS-PCR.

## Statistical analysis

DNA sequences were manually examined using BioEdit version 7.2.3.0 (Hall 1999) and aligned using ClustalW (Thompson *et al.* 2002). Population genetic parameters were assessed with DnaSP version 6.12.03 (Rozas *et al.* 2003). A haplotype network was constructed using the TCS program (Clement *et al.* 2000), and a maximum likelihood phylogenetic tree was constructed using MEGA X (Kumar *et al.* 2018).

Statistical analyses were performed with Prism 8 (GraphPad Software, San Diego, California USA). Alpha values for significance were set at $P < 0.05$, and all confidence intervals (CI) were reported at 95%. For data generated from metabolism assays and contact bioassays with transgenic *D. melanogaster* flies, Student's *t*-test was used to test for significance. Fisher's exact test was used to determine the significance of any observed differences in proportions within genotype contingency tables. Statistical significance was denoted by asterisks: *$P < 0.05$, **$P < 0.01$, and ***$P < 0.001$.

**Table 1.** Africa-wide polymorphism parameters of *CYP9K1*.

| Sample | $n$ | S | h | $H_d$ | Syn | Nsyn | Pi | D | D* |
|---|---|---|---|---|---|---|---|---|---|
| FUMOZ | 15 | 1 | 2 | 0.24 | 1 | 0 | 0.00015 | $-0.39^{ns}$ | $0.70^{ns}$ |
| FANG | 14 | 13 | 6 | 0.78 | 8 | 5 | 0.00221 | $-0.51^{ns}$ | $-1.20^{ns}$ |
| Cameroon 2014 | 40 | 107 | 38 | 0.99 | / | / | 0.01346 | $-0.74^{ns}$ | $-1.09^{ns}$ |
| Cameroon 2020 | 13 | 2 | 3 | 0.29 | 2 | 00 | 0.00019 | $-1.46^{ns}$ | $-1.77^{ns}$ |
| Uganda 2014 | 40 | 29 | 5 | 0.19 | 16 | 19 | 0.00113 | $-2.51^{***}$ | $-3.81^{**}$ |
| Uganda 2020 | 14 | 1 | 2 | 0.14 | 1 | 00 | 0.00009 | $-1.15^{ns}$ | $-1.39^{ns}$ |
| Malawi 2014 | 40 | 39 | 29 | 0.97 | 37 | 2 | 0.00932 | $2.24^{ns}$ | $1.06^{ns}$ |
| Malawi 2020 | 13 | 33 | 11 | 0.97 | 31 | 2 | 0.0082 | $1.07^{ns}$ | $1.19^{ns}$ |
| Ghana 2020 | 11 | 23 | 7 | 0.87 | 21 | 2 | 0.0046 | $-0.25^{ns}$ | $1.02^{ns}$ |
| Total All | 80 | 55 | 29 | 0.87 | 47 | **9** | 0.0098 | $1.32^{ns}$ | $0.08^{ns}$ |

Abbreviations: *n*, number of sequences (2n); S, number of polymorphic sites; Syn, synonymous mutations; *h*, haplotype; $H_d$, haplotype diversity; Nsyn, Non-synonymous mutations; Pi, nucleotide diversity; *D* and *D**, Tajima's and Fu and Li's statistics; ns, not significant.

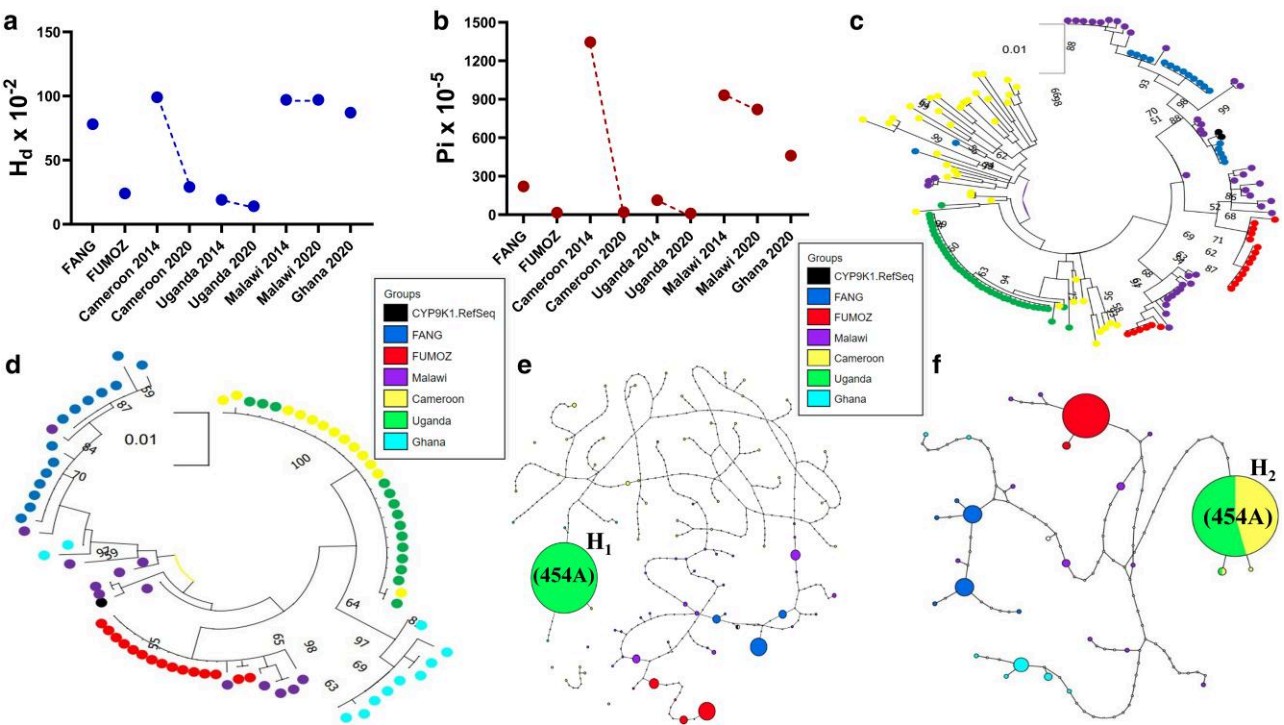

**Fig. 1.** Directional selection of 454A-*CYP9K1* dominant allele in *An. funestus* across Africa. Spatiotemporal variation of (a) Haplotype diversity ($H_d$) and (b) nucleotide diversity of *CYP9K1* showing signatures of directional selection in Uganda and Cameroon between 2014 and 2020. c) Maximum likelihood tree of *CYP9K1* coding length across Africa using 2014 samples. d) Maximum likelihood tree of *CYP9K1* coding length across Africa using 2020 samples. e) Haplotype network representation of *CYP9K1* coding across Africa using 2014 samples showing dominant haplotype shared between Uganda samples. f) Haplotype network representation of *CYP9K1* coding across Africa using 2020 samples showing dominant haplotype shared between Cameroon and Uganda in 2020.

QGIS version 3.14.16 (www.qgis.org) was used to generate a map depicting the percentage frequency of various genotypes of the G454A-*CYP9K1* mutation across different genotyping sites in Africa.

## Results

### Africa-wide temporal genetic variability analysis detected a dominant and rapidly spreading 454A-*CYP9K1* haplotype

Genetic variation analysis of *CYP9K1* from 2014 samples revealed reduced diversity in Uganda and high diversity in Cameroon (Mibellon), with respectively 35 vs 123 substitution sites (S) and 0.19 vs 0.99 haplotype diversities ($h_d$) (Table 1, Fig. 1a). Interestingly, by 2020 Cameroon samples exhibited reduced diversity comparable to Uganda samples in 2014 and 2020 ($h_d$ = 0.295, Pi = 0.00019 and $h_d$ = 0.143, Pi = 0.00009, respectively, for Cameroon

and Uganda in 2020). This is in contrast with the samples from Malawi ($h_d$ = 0.974, Pi = 0.00820) and Ghana ($h_d$ = 0.873, Pi = 0.0046), both of which registered higher diversities (Table 1, Fig. 1, a and b). Uganda 2014 samples, clustered together, forming a dominant clade not seen in Cameroon and Malawi (Fig. 1c). However, analysis of 2020 samples resulted in Cameroonian samples joining the Uganda haplotype cluster, distinct from other African regions, e.g. Southern Africa (Malawi) and West Africa (Ghana) (Fig. 1d).

Furthermore, haplotype network analysis of 2014 samples revealed a dominant *CYP9K1* haplotype ($H_1$) present at a frequency of 90% in Uganda and absent in Malawi and Cameroon. In 2014, Uganda registered a total of 5 haplotypes vs 38 and 23, respectively in Cameroon and Malawi, both of which formed singleton haplotypes (Fig. 1e). In 2020, Cameroon and Uganda samples both shared the same dominant haplotype ($H_2$) previously detected in Uganda in 2014 (Fig. 1f). By 2020 the $H_2$ haplotype had become

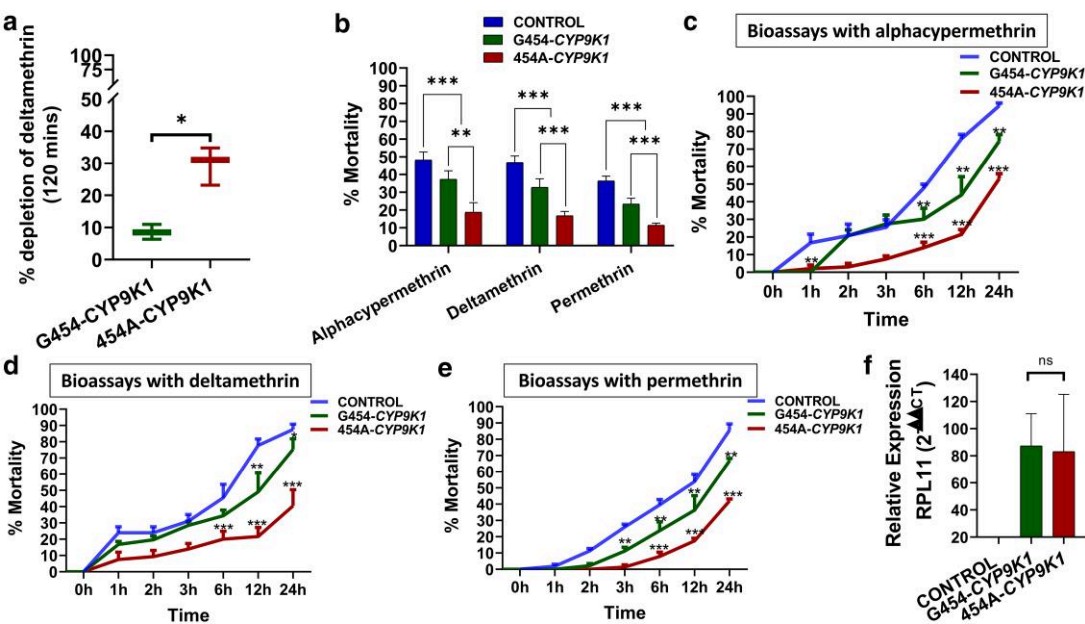

**Fig. 2.** Mutant-type 454A-*CYP9K1* allele drives pyrethroid resistance through in vitro and in vivo experiments. a) Percentage depletion of 20 μM deltamethrin by recombinant *CYP9K1* membranes. Horizontal bar represents the mean percentage depletion, and the error bars represent the 95% confidence intervals of the mean. The *P*-value is calculated using a Welch's *t*-test. Results are an average of 3 replicates compared between the 2 groups. b) Average percentage mortality of *CYP9K1* allelic variants 24 h after exposure to pyrethroid insecticide; comparative mortalities of F$_1$ transgenic progenies of crosses between Actin5C-GAL4 and *UAS-CYP9K1* on exposure to (c) 0.0007% alphacypermethrin; d) 0.2% deltamethrin and (e) 4% permethrin; f) Comparative relative expression of G454-*CYP9K1* and 454A-*CYP9K1* in *Drosophila* flies. Data are represented as mean ± SEM: *$P$ < 0.05, **$P$ < 0.01 and ***$P$ < 0.001.

predominant in Cameroon with a frequency nearing fixation at 84.6%, while remaining absent in Southern Africa (Malawi) and West Africa (Ghana). Samples of 2020 comprised a total of 29 haplotypes, with Cameroon and Uganda respectively registering 3 and 2 haplotypes compared to 11, 7, and 6, respectively, for Malawi, Ghana, and FANG (Table 1).

Sequence analysis identified a point mutation guanine (G) to cytosine (C) nucleotides substitution at position 1361, leading to the replacement of glycine (G) with alanine (A) on codon 454 (Supplementary Fig. 1, a and b). This mutation was fixed in Uganda (40/40) and at low frequency in Cameroon (9/40) and Malawi (13/40) in 2014. However, by 2020 the mutation was fixed in FUMOZ (15/15), Uganda (14/14, 100%), and Cameroon (13/13, 100%), but remained at a low frequency in Malawi (4/13, 30.7%) and was completely absent in both Ghana (0/11, 0%) and FANG [0/14, 0% (Supplementary Fig. 2, a and b)]. Another point mutation, a guanine to adenine (A) nucleotides substitution at position 979, resulting replacement of valine (V) with isoleucine (I) on codon 327 was detected at high frequency (9/11, 81.8%) in samples from Ghana (West Africa), but was absent in samples from Cameroon, Malawi, and Uganda (Supplementary Fig. 2, a and b).

## Comparative in vitro and in vivo experiments confirmed the mutant 454A-*CYP9K1* allele drives pyrethroid resistance

### Comparative in vitro assessment of metabolic activity of *CYP9K1* allelic variants

Co-expression of recombinant *CYP9K1* with *CPR* revealed optimal expression between 22–24 h post-induction with 1 mM IPTG and 0.5 mM δ-ALA. Both alleles of *CYP9K1*, when complexed with standard P450 carbon monoxide (CO), generated similar difference spectra with absorbance peaks at 450 nm wavelength (Supplementary Fig. 3, a and b). The recombinant proteins

exhibited comparable concentrations of 3.27 μM for the 454A-*CYP9K1* allele and 3.67 μM for the G454-*CYP9K1* allele.

Preliminary metabolism assays demonstrated that the recombinant 454A-*CYP9K1* allele depleted a significantly higher amount (29.7%) of deltamethrin compared to the G454-*CYP9K1* allele, which depleted only 8.5% of this type II pyrethroid (Fig. 2a). The depletion of deltamethrin was 21.10% ± 3.64 higher in the mutant-type allele than in the wild-type allele (*t*-test: *t* = 5.73; df = 2.59; *P* = 0.01). However, as previously established in previous studies, both alleles did not deplete permethrin, suggesting a lack of metabolic activity towards this type I pyrethroid (Hearn *et al.* 2022).

### The 454A-CYP9K1 *allele confers higher pyrethroid resistance to* D. melanogaster *flies*

Insecticide bioassays revealed significant differences in the average percentage mortalities for alphacypermethrin (*n* = 120), deltamethrin (*n* = 120), and permethrin (*n* = 120). The average respective mortalities were 18.26% ± 5.33 vs 37.27% ± 4.73 (P < 0.0004), 16.83% ± 2.41 vs 32.77% ± 4.76 (P < 0.0006), and 11.44% ± 1.15 vs 23.36% ± 3.24 (P < 0.0006) for the flies expressing 454A-*CYP9K1* construct compared to those expressing the wild-type G454-*CYP9K1* construct (Fig. 2b).

***Bioassays with alphacypermethrin:*** Time-course assay revealed that the 454A-*CYP9K1* transgenic flies were more resistant to alphacypermethrin than the wild-type transgenic flies, with mortalities of less than 10% vs 20% in the first 2 h (P < 0.0001), 20% vs 30% in 6 h (P = 0.002), and 30% vs 45% after 12 h (P = 0.0066), respectively (Fig. 2c).

***Bioassays with deltamethrin:*** Deltamethrin exposure revealed lower mortality rates in flies expressing the 454A-*CYP9K1* allele, compared with the flies expressing the G454-*CYP9K1* allele, at 10% vs 20% in 2 h (P = 0.0057), 20% vs 35% in 6 h (P = 0.0038), and 30% vs 50% after 12 h (P = 0.0054), respectively (Fig. 2d).

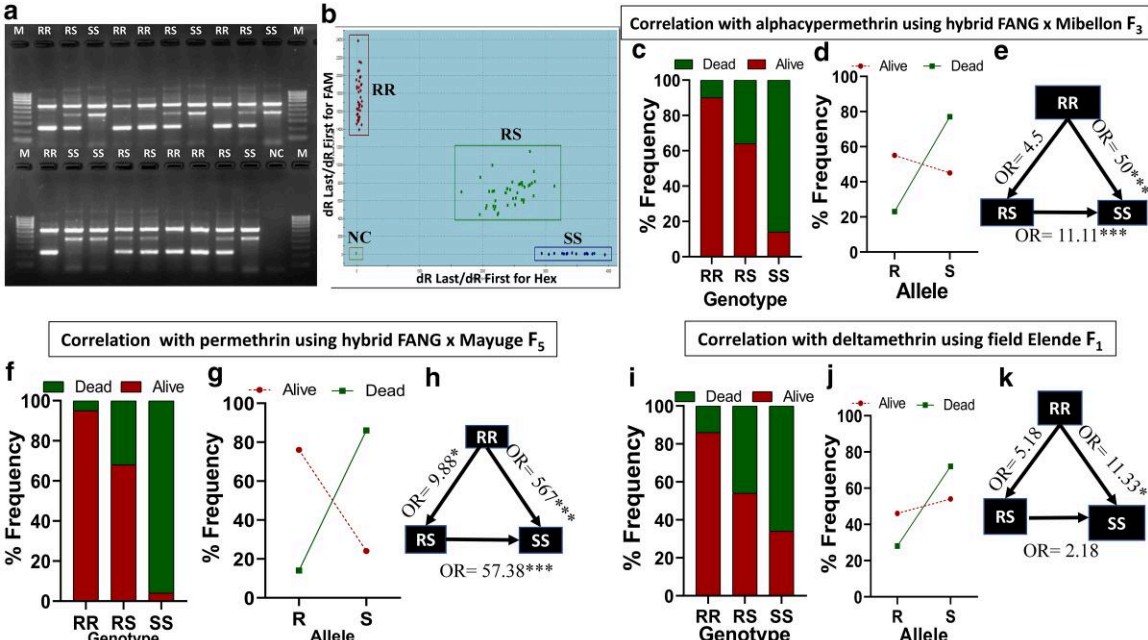

**Fig. 3.** DNA-based diagnostic assay for G454A-*CYP9K1* marker is a robust tool to detect pyrethroid resistance. a) Agarose gel image of G454A-*CYP9K1* AS-PCR showing homozygous resistant (RR, with band size 216 bp and a common band 639 bp), heterozygous (RS, band sizes 216 and 434 bp with at common band 639 bp), homozygous susceptible (SS, with band size 434 bp plus a common band 639 bp) genotypes, b) Dual color scatter plot of *CYP9K1*-probe-based locked-nucleic acid (LNA) assay showing RR genotypes clustered on the y-axis and SS genotypes clustered on the x-axis, RS genotypes between the x-axis and y-axis and negative controls (NC), c) G454A-*CYP9K1* percentage genotype frequency distribution in FANG × Mibellon $F_3$ alphacypermethrin alive and dead female mosquitoes. d) Percentage allele frequency distribution FANG × Mibellon $F_3$ alphacypermethrin alive and dead female mosquitoes. e) Estimation of odds ratio (OR) and associated significance between different *CYP9K1* genotype in alphacypermethrin alive and dead FANG × Mibellon $F_3$ female mosquitoes. The arrow within the triangle indicates the direction of OR estimation and ORs are given with asterisks indicating level of significance. (f), (g), (h) are respectively the same as (c), (d), (e) using FANG × Mayuge $F_5$ permethrin alive and dead female mosquitoes. i) G454A-*CYP9K1* genotype frequency distribution in Elende 2022 $F_1$ deltamethrin alive and dead female mosquitoes. j) Allelic frequency distribution in Elende 2022 $F_1$ alive and dead to deltamethrin exposure. k) Estimation of odds ratio (OR) and associated significance between different *CYP9K1* genotype in Elende 2022 $F_1$ deltamethrin alive and Dead.

**Bioassays with permethrin:** Exposure to permethrin resulted in a similar pattern, with less than 10% vs 22% mortalities in 454A-*CYP9K1* allele flies compared with the G454-*CYP9K1* allele flies in the first 6 h ($P = 0.0049$), less than 20% vs 32% in 12 h ($P = 0.0219$) and 40% vs 62% at 24 h [$P < 0.0001$ (Fig. 2e)].

Both alleles of *CYP9K1* were overexpressed in the *GAL4-UAS-CYP9K1* $F_1$ progenies, while no expression was observed in the control (flies generated from the crossing between the UAS line without the *CYP9K1* gene and the Act5C-GAL4 driver line) (Fig. 2f). No significant difference ($t$-test = 0.14, df = 3.15) was observed in the expression levels between the G454-*CYP9K1* flies (fold change = 87.3) and the 454A-*CYP9K1* flies (fold change = 83.17).

## DNA-based diagnostic assay to detect 454A-*CYP9K1* mutation

To detect and track resistance driven by the 454A-*CYP9K1* allele, 2 PCR-based assays were successfully designed: an allele-specific PCR (AS-PCR) and a probe-based locked nucleic acid (LNA) assay (Fig. 3, a and b). These assays target the glycine (1361-G**G**A) to alanine (1361-G**C**A) mutation in codon 454. Applying this AS-PCR assay on genomic DNA samples extracted from Uganda female *An. funestus* mosquitoes (Mayuge $F_0$ 2021) revealed that the 454A/A-*CYP9K1* (RR) mutant-type genotype was fixed (100% frequency). All the samples exhibited a mutant-type band at 216 bp and a common band at 639 bp on gel image (Supplementary Fig. 4a), corresponding to the RR genotype. Conversely, all laboratory-susceptible samples (FANG) were genotyped as homozygous wild-type G/G454-*CYP9K1*

genotype (SS), with the wild-type band size at 434 bp plus a common band at 639 bp on gel image (Supplementary Fig. 4a).

This was confirmed with an LNA assay, where all Uganda female $F_0$ samples clustered together on the y-axis on the LNA dual scatter plot (Supplementary Fig. 4b), corresponding to the homozygous RR genotype. On the contrary, the laboratory-susceptible samples (FANG) clustered together on the x-axis in the LNA dual scatter plot (Supplementary Fig. 4b), corresponding to the homozygous SS genotype.

To achieve more robust genotype segregation, hybrid strains from crosses between the susceptible lab strain FANG and field Mayuge mosquitoes reared to $F_5$ generation were genotyped, depicting better segregation of genotypes (Fig. 3, a and b). Samples harboring the heterozygous genotype (RS) produced all 3 bands (639, 434, and 216 bp) on the gel image, and in the LNA dual scattered plot, the heterozygous genotype (RS) clustered in between the y-axis and x-axis (Fig. 3, a and b).

### *The 454A-CYP9K1 mutation strongly correlates with pyrethroid resistance phenotype using hybrid* An. funestus

Genotyping of hybrid, $F_3$ generation mosquitoes generated from crosses between FANG and Mibellon populations revealed that those that survived 1 h to 0.05% alphacypermethrin exposure were mainly homozygote resistant (9/44) and heterozygotes (31/44), translating into 55% resistant allele (R) frequency, with only 4 homozygote susceptible samples (Fig. 3c). In contrast, among dead mosquitoes after 10 mins exposure to alphacypermethrin were predominantly homozygote susceptible (25/44) and

heterozygotes (18/44) with only one homozygote resistant (Fig. 3c), consequently harboring 77% of the susceptible (S) allele (Fig. 3d). Increased alphacypermethrin survival was observed in RS vs SS mosquito samples comparison (OR = 11.11; CI = 3.31–32.28; $P$ < 0.0001). An additive effect was recorded for the RR vs SS genotypes comparison (OR = 50; CI = 5.23–566.4; $P$ < 0.0001, Fisher's exact test) (Fig. 3e, Supplementary Table 4).

Similar findings were obtained using hybrids from crossings between FANG and Mayuge at the $F_5$ generation. Mosquitoes that survived 1 h of exposure to 0.75% permethrin were predominantly RR (22/40) and RS (17/40), translating to 77% resistant allele (R) frequency, with only one SS mosquito sample (Fig. 3f). Dead mosquitoes after 30-min exposure to permethrin were mainly SS (29/40) and RS (9/40) with only a single RR genotype (Fig. 3f), bearing 85% of the susceptible (S) allele (Fig. 3g). An increased ability to survive permethrin exposure was recorded in RS vs SS (OR = 57.38; CI = 7.47–617.6; $P$ < 0.001) and even more pronounced in RR vs SS mosquitoes (OR = 567; CI = 34.03–5831; $P$ < 0.0001) (Fig. 3h, Supplementary Table 4).

### The 454A-CYP9K1 *allele (R) correlates with resistance in field mosquitoes*

WHO tube bioassays conducted using female mosquitoes from Elende revealed mortality rates of 34.4% ± 2.8, 70.21% ± 2.5 and 59.55% ± 1.7 after 1-h exposure to 0.05% alphacypermethrin, 0.05% deltamethrin, and 0.75% permethrin, respectively.

Higher frequencies of RR (5/40) and RS (28/40) genotypes were recorded in field mosquito samples that survived 1-h alphacypermethrin exposure (Supplementary Fig. 5a). The alphacypermethrin-dead mosquitoes were predominantly SS (18/32) and RS (12/32), with only 2 mosquitoes being RR (Supplementary Fig. 5a), resulting in a high frequency (71.4%) of the susceptible (S) allele (Supplementary Fig. 5b). A significant association was observed between alphacypermethrin resistance and the 454A-*CYP9K1* mutation when comparing RR vs SS (OR = 7.1; CI: 2.2–22.64; $P$ = 0.0006, Fisher's exact test) and RS vs SS [OR = 6; CI: 1.897–17.42; $P$ < 0.001 Fisher's exact test (Supplementary Fig. 5c, Supplementary Table 4)].

The field mosquitoes which samples that survived 1-h deltamethrin exposure harbored higher proportions of RR (6/37) and RS (24/37) genotypes, while deltamethrin-dead mosquitoes were predominantly SS (17/37) and RS (19/37) with a single RR sample (Fig. 3i). Consequently, dead samples harbored a high frequency (72%) of the susceptible (S) allele (Fig. 3j). Mosquitoes bearing RR genotype significantly survived deltamethrin exposure than SS mosquitoes [OR = 11.33; CI: 1.25–134.1; $P$ = 0.03, Fisher's exact test (Fig. 3k, Supplementary Table 4)].

A similar pattern was observed for field mosquitoes that survived permethrin, with higher proportions of RR (5/32) and RS (20/32) genotypes compared to dead mosquitoes, from which 17/32 and 14/32 harbored the SS and RS genotypes, respectively (Supplementary Fig. 5d). The wild (S) allele was more associated with susceptibility to permethrin, with 75% of this allele found in the dead mosquitoes (Supplementary Fig. 5e). Field mosquitoes with the RR genotype were significantly more capable of surviving permethrin compared with the SS genotype (OR = 12.14; CI = 1.55–49.4; $P$ < 0.025, Fisher's exact test) (Supplementary Fig. 5f).

### Correlation between the 454A-CYP9K1 *mutation and resistance toward the LLINs using cone bioassays*

To assess the impact of 454A-*CYP9K1* mutation on the efficacy of LLLINs, cone bioassays were performed. Test performed with different LLINs using Elende $F_1$ female *An. funestus* mosquitoes revealed low mortality rates of 8.3% ± 2.3, 10.53% ± 1.5, 13.56% ± 3.1, 18.33% ± 2.2, and 10.23% ± 2.7, respectively, with the conventional pyrethroid-only nets, PermaNet 3.0 side panel, PermaNet 2.0, Royal sentry, Olyset, and DuraNet. However, for the PBO-containing nets, PermaNet 3.0 top and Olyset plus 100% mortalities were obtained, highlighting the greater insecticidal efficacy of these combination bed nets.

Genotyping of the above mosquitoes tested with a Royal sentry net revealed high proportions of RR (8/36) and RS (21/36) genotypes in the mosquitoes which survived exposure, while only 7/36 harbored the SS genotype. The dead mosquitoes after were predominantly SS (20/35) and RS (14/35) with a single RR genotype (Fig. 4a), resulting in a high frequency (77%) of the susceptible (S) allele (Fig. 4b). Increase survivorship was observed in RS and RR when compared to SS mosquitoes [RS vs SS, OR = 3.5; CI = 1.18–9.51; $P$ < 0.02 and RR vs SS, OR = 20; CI = 2.57–230.3; $P$ < 0.0023, Fisher's exact test (Fig. 4c)].

Similar findings were observed with the Olyset, where RR (5/30) and RS (17/30) genotypes were more prevalent in surviving mosquitoes that survived. Dead samples were primarily SS (23/32) and RS (8/32) with only one RR (Fig. 4d), resulting in a high frequency (84%) of the susceptible (S) allele among the dead mosquitoes (Fig. 4e). A positive correlation was observed between the 454A-*CYP9K1* mutation and Olyset resistance when comparing RS vs SS genotypes (OR = 4.31; CI = 1.43–12.79; $P$ < 0.014) and RR vs SS [OR = 11.5; CI = 1.48–140.0; $P$ < 0.023 (Fig. 4f, Supplementary Table 5)].

## Evaluation of the operational impact of 454A-*CYP9K1* mutation on the efficacy of LLINs and its spatiotemporal distribution

### Correlation between the 454A-CYP9K1 *mutation and resistance toward net using experimental hut trials*

Mosquitoes caught during the trials in experimental huts were genotyped for the 454A mutation. In total, Genotype distribution analysis revealed most of the RR (7/50) genotype was found in mosquitoes alive in huts treated with Royal sentry (alphacypermethrin), 23/50 and 20/50 were RS and SS genotypes, respectively. Dead mosquitoes collected in huts treated with Royal sentry were predominantly SS (24/49) and RS (23/49), with 2/49 RR samples (Fig. 4g), translating to a high frequency (73%) of the susceptible (S) allele (Fig. 4h). Homozygous resistant (RR) mosquitoes were 3.8 times more likely to survive exposure to Royal sentry than homozygous susceptible (SS) mosquitoes [OR = 3.81; CI = 0.82–19.29; $P$ = 0.14, Fisher's exact test (Fig. 4i)].

A high proportion of RS (55/73) mosquitoes were alive after exposure to huts where volunteers slept in PBO-containing net (PermaNet 3.0), with the rest being RR (5/73) and SS [13/73 (Fig. 4j)]. Dead mosquitoes from huts treated with this net were predominantly SS (20/50) and RS (25/50), with 5 RR, hence bearing higher frequency (65%) of the susceptible allele (Fig. 4k). Heterozygote (RS) mosquitoes significantly survived exposure to this net compared to SS mosquitoes [OR = 3.43; CI = 1.5–7.64; $P$ = 0.0057, Fisher's exact test (Fig. 4l, Supplementary Table 5)].

### Africa-wide spatiotemporal distribution of 454A-CYP9K1 *mutation*

Analysis of percentage genotype distribution in field samples ($n$ = 40) from West Africa revealed a complete absence of the RR genotype, with most samples harboring the SS genotype, e.g. in Ghana (83%) and Benin (97%) from 2021 collection (Fig. 5a). Moderate frequencies of the RR genotype were detected in southern Africa, e.g. in Zambia (28%) from 2014, Mozambique (10%) from 2020, and Malawi (27%) from 2021 collections. In East Africa, the RR mosquitoes

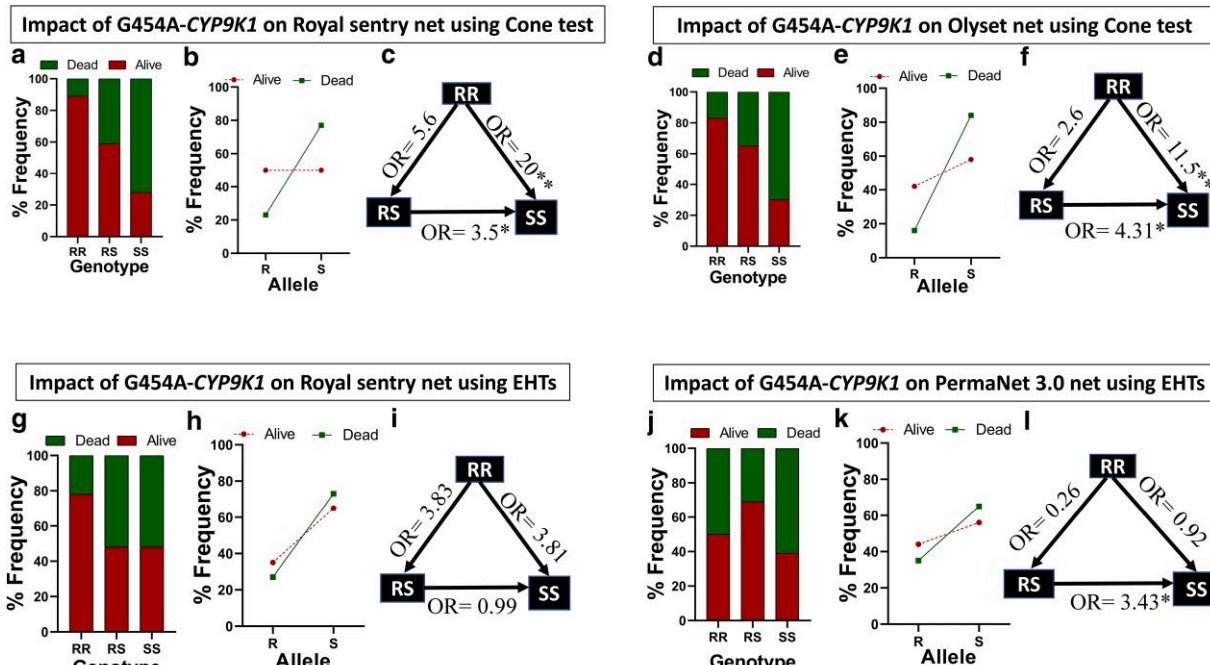

**Fig. 4.** Mutant-type 454A-CYP9K1 allele reduces the efficacy of LLINs using cone test and EHTs samples. a) *CYP9K1* percentage Genotype frequency distributions in Elende $F_1$ alive and dead to Royal sentry using cone test. b) *CYP9K1* percentage allele frequency distributions in Elende $F_1$ alive and dead to Royal sentry using cone test. c) Estimation of odds ratio (OR) and associated significance between different *CYP9K1* genotypes and the ability to survive exposure to Royal sentry using cone test. The arrow within the triangle indicates the direction of OR estimation and ORs are given with asterisks indicating level of significance. (d), (e), (f) are respectively the same as (a), (b), (c) using Elende 2022 $F_1$ alive and Dead to Olyset exposure. g) *CYP9K1* percentage Genotype frequency distributions in Elende 2022 $F_0$ free-flying alive and dead to Royal sentry using EHT. h) *CYP9K1* percentage allele frequency distributions in Elende 2022 F0 free-flying alive and dead to Royal sentry using EHT. i) Estimation of odds ratio (OR) and associated significance between different *CYP9K1* genotypes and the ability to survive exposure to Royal sentry using EHT. The arrow within the triangle indicates the direction of OR estimation and ORs are given with asterisks indicating level of significance. (j), (k), and (l) are respectively the same as (g), (h), (i) using Elende 2022 $F_0$ free-flying alive and Dead to PermaNet 3.0 exposure using EHTs.

were present at a moderate frequency in Tanzania (58%) from 2018 collection and fixed (100%) in Uganda populations collected in 2021. The RR genotype was very high (90%) in mosquitoes from Mibellon (Cameroon, Central Africa), which were collected in 2021 (Fig. 5a). However, moderate frequencies of the RR genotype were detected in other regions/localities in Cameroon, e.g. Penja (39%) in 2021 and Elende (27%) in 2023, while low frequencies were seen in Tibati (5%) and in Gounougou (3%) collections from 2021 (Fig. 5a).

High frequencies of the resistant allele were observed in East Africa (Uganda since 2010, Tanzania in 2018) and some Central African regions (Mibellon in Cameroon), while moderate frequencies of the alleles were observed in Southern Africa (e.g. Malawi in 2014 and 2021 and Mozambique in 2014 and 2020) and a high frequency of susceptible (S) allele were seen in West Africa [Ghana in 2021 collection (Fig. 5b)].

Temporal analysis of East African samples revealed an increase in the frequency of the RR genotype in Uganda mosquitoes (from 80% in 2010 to 93% in 2016 and 100% in 2021) (Fig. 5c) and Tanzania (from 17% in 2014 to 58% in 2018). The frequency of resistance allele (R) also increased between 2010 (91%), 2016 (95%), and 2021 (100%) in Uganda (Fig. 5d) and in Tanzania between 2014 (46%) and 2018 (68%).

In Central Africa, the frequencies of both the RR genotype and the R allele increased in mosquito samples from Mibellon district ($n = 40$) between 2014 (10 and 28%), 2018 (25 and 62%), and 2021 (90 and 94%) (Fig. 5, e and f) and in Elende district ($n = 50$) between 2019 (8 and 28.5%), 2021 (15 and 35%), and 2023 (27 and 53%) (Fig. 5, g and h).

In Southern Africa ($n = 44$), the frequency of the RR genotype increased between 2014 (15%) and 2021 (27%) in Malawi (Fig. 5i) and between 2016 (6%) and 2020 (10%) in Mozambique. However, the frequency of the resistant allele (R) remained relatively stable in Malawi between 2014 (42%) and 2021 (43%) (Fig. 5j) and in Mozambique between 2016 (34%) and 2020 (33%).

## Discussion

Dissecting the genetic bases of insecticide resistance is imperative for effective control of public health vectors, including the malaria-transmitting mosquitoes. Molecular markers of resistance are valuable tools that allow tracking the spread and evolution of resistance, enhancing planning and decision-making which improve health and wellbeing. This study reported a field-to-laboratory-to-field to determine the role of a major P450 *CYP9K1* gene in insecticide resistance in the major malaria vector *An. funestus* and established new diagnostics for tracking insecticide resistance in real-time in this species. Applying this assay for metabolic resistance surveillance across African regions confirmed the resistant 454A-CYP9K1 allele is under strong directional selection in Central and East Africa, and is strongly associated with pyrethroid resistance in these regions, which can reduce the efficacy of LLINs.

### The 454A-*CYP9K1* haplotype selected in Eastern Africa has spread to Central Africa

The reduced genetic polymorphism of the *CYP9K1* gene, initially observed only in Uganda in 2014, has now spread to Cameroon.

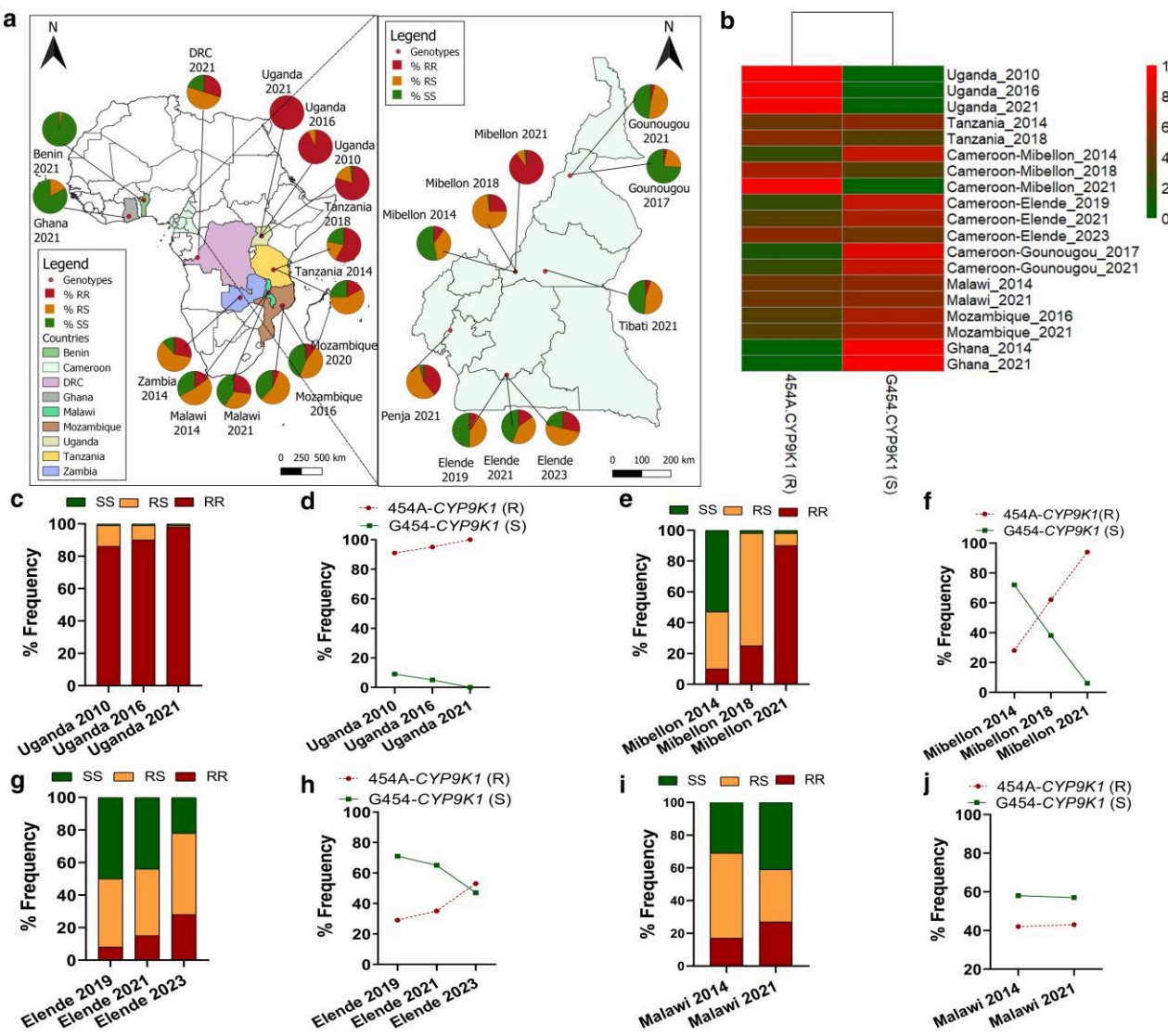

**Fig. 5.** Spatiotemporal distribution of G454A-*CYP9K1* marker across Africa. a) Africa-wide and Cameroon-wide spatiotemporal distribution of the percentage frequency of G454A-*CYP9K1* genotypes in *An. funestus* mosquito populations, b) Heat map showing geographical and temporal evolution of percentage frequency of *CYP9K1* alleles across Africa, showing increase in the frequency of the resistant (R) allele with time in Uganda, Tanzania and Cameroon, no major change in frequencies of both alleles of *CYP9K1* with time in Malawi and Mozambique and very low frequency of the resistant (R) allele in Ghana (c) Temporal genotype distribution of the G454A-*CYP9K1* marker in Uganda, showing fixation of the RR genotype in 2021, d) Temporal evolution of the frequency of *CYP9K1* alleles in Uganda, showing fixation of resistant (R) allele and reduction to zero of the susceptible (S) allele (e) and (f) are respectively same as (c) and (d) in Cameroon, Mibellon showing an increase in frequency of the RR genotype and resistant (R) allele and decrease in the frequency of the susceptible (S) allele with time. (g) and (h) are, respectively, same as (c) and (d) in Cameroon, Elende showing an increase in frequency of the RR genotype and resistant (R) allele and a decrease in the frequency of the susceptible (S) allele with time. (i) and (j) are, respectively, the same as (c) and (d) in Malawi showing an increase in the frequencies of both the RR and the SS genotypes with time (i) and no major change in the frequencies of both the resistant (R) and susceptible (S) alleles in Malawi between 2014 and 2021 (j).

This rapid decline in diversity, evident in Cameroon between 2014 and 2020, is likely attributed to strong selective pressure exerted by insecticide-treated nets. This could have contributed to the directional selection of the 454A-*CYP9K1* allele in these populations. The fact that the same haplotype is predominant in 2 regions of Africa suggests that this mutant allele might have spread across the Equatorial zone from East to West Africa. This pattern mirrors recent findings of the 4.3-kb transposon-based pyrethroid-resistant allele in *An. funestus* (Mugenzi *et al.* 2024), indicating substantial gene flow between Uganda and Cameroon (Barnes *et al.* 2017; Weedall *et al.* 2020). Conversely, the high genetic polymorphism of *CYP9K1* in Malawi (Southern Africa), and laboratory mosquito strain FANG highlights the limited role of the resistance 454A allele in these regions. Previous studies have

documented rapid selection of resistance, as exemplified by the increased frequency of the *An. coluzzii CYP9K1* haplotype in Bioko Island, Equatorial Guinea, between 2011 and 2015 following the reintroduction of pyrethroid-based control interventions (Vontas *et al.* 2018), and in Selinkenyi, Mali, before and after 2006 bed net scale-up (Main *et al.* 2015). These findings underscore the dynamic nature of insecticide resistance evolution in mosquito populations.

Analyses revealed a single, fixed *CYP9K1* haplotype (454A) in mosquito samples from Uganda and Cameroon, underscoring the alarming rate of the spread of this allele from Eastern to Central Africa. Studies on other *An. funestus* cytochrome P450 monooxygenases *CYP6P9a/b*, *CYP6Z1*, *CYP6Z3* and the glutathione-S-transferase epsilon 2 loci (*GSTe2*) genes have highlighted selection of resistant alleles

for each of these genes (Wondji *et al.* 2009; Irving *et al.* 2012; Riveron *et al.* 2013; Riveron, Cristina, *et al.* 2014; Riveron, Ibrahim, *et al.* 2014; Ibrahim *et al.* 2015). While studies on other cytochrome P450 genes have identified resistant alleles, genetic diversity analyses of *CYP6M2* and *CYP6P4* in *An. coluzzii* and *An. gambiae* s.s. and *An. funestus CYP6Z1*, *CYP6Z3* and *CYP325A* genes have shown no dominant allele selection despite their role in pyrethroid resistance (Ibrahim *et al.* 2015; Wagah *et al.* 2021; Wamba *et al.* 2021; Fotso-Toguem *et al.* 2022). These findings highlight the complex dynamic of insecticide resistance evolution in mosquito populations.

## The highly selected 454A-*CYP9K1* allele is more efficient in metabolizing pyrethroid insecticide

Comparative metabolism assays with the recombinant *CYP9K1* alleles revealed that the mutant-type 454A-*CYP9K1* allele exhibits higher metabolic activity towards deltamethrin, while no activity was observed towards permethrin. This suggests that allelic variation significantly influences the metabolic activity of the *CYP9K1* gene and contributes to increased insecticide resistance. Previous research has shown that the 454A-*CYP9K1* allele, coupled with *An. gambiae CPR,* primarily metabolizes deltamethrin (Vontas *et al.* 2018; Hearn *et al.* 2022). The deltamethrin percentage depletion (~30%) obtained from this study was similar to what was reported in a previous study (Vontas *et al.* 2018). The lack of metabolic activity toward permethrin was reported in several studies using this P450 (Hearn *et al.* 2022). This observation is similar to what was reported in other species; for example, *An. arabiensis CYP6P4* gene was reported to metabolize permethrin, but exhibited no metabolic activity toward deltamethrin (Ibrahim, Ndula, *et al.* 2016; Ibrahim, Riveron, *et al.* 2016). Several studies have demonstrated that allelic variation is a key factor driving differences in metabolic efficiencies for resistant genes such as *CYP6P9a/b, CYP325A,* and *GSTe2* (Riveron, Cristina, *et al.* 2014; Riveron, Ibrahim, *et al.* 2014; Ibrahim *et al.* 2015; Wamba *et al.* 2021; Mugenzi *et al.* 2023). Functional validation of the brown planthopper, *Nilaparvata lugens CYP6ER1* gene further highlights the role of single mutations in enhancing the metabolic activity of this detoxification enzyme, driving resistance to imidacloprid (Zimmer *et al.* 2018) and cross-resistance to ethiprole (Duarte *et al.* 2022) in this insect pest of crops.

## Overexpression and allelic variation of *CYP9K1* confer higher pyrethroid resistance

This study demonstrated that while both *CYP9K1* alleles confer resistance, flies expressing the mutant-type allele exhibit higher resistance levels compared to those expressing the wild-type allele. This underscores the significant contribution of allelic variation (G454A) in addition to overexpression in conferring elevated resistance in the flies carrying the mutant-type 454A-*CYP9K1* allele. Previous studies have shown that transgenic *Drosophila* flies independently expressing the resistant alleles of the duplicated *CYP6P9a* and *CYP6P9b* genes exhibit higher resistance to pyrethroids (Riveron *et al.* 2013; Ibrahim *et al.* 2015) and cross-resistance to carbamate insecticide (Mugenzi *et al.* 2023). These findings align with those of (Riveron, Cristina, *et al.* 2014; Riveron, Ibrahim, *et al.* 2014) who reported that a single mutation of L119F in the *GSTe2* gene from *An. funestus* populations is responsible for high resistance to both DDT and permethrin (Riveron, Cristina, *et al.* 2014; Riveron, Ibrahim, *et al.* 2014; Riveron *et al.* 2017). In other species, similar patterns have been observed. For example, transgenic expression of brain-specific P450 *CYPBQ9* from *Tribolium castaneum* in *D. melanogaster* resulted in increased tolerance and reduced mortality to deltamethrin insecticide compared to control flies (Zhu *et al.* 2010). Transgenic expression of other cytochrome P450s like *CYP6G1,*

*CYP6G2,* and *CYP12D1* independently in *D. melanogaster* flies conferred increased survival to at least one class of insecticide among DDT, nitenpyram dicyclanil, and diazinon compared to control flies (Daborn *et al.* 2007). Furthermore, transgenic expression of allelic variants of *Nilaparvata lugens CYP6ER1* in *D. melanogaster* revealed significantly increased survival to imidacloprid and ethiprole in experimental flies expressing the mutant alleles compared to those expressing the wild alleles (Zimmer *et al.* 2018; Duarte *et al.* 2022).

## The novel 454A-*CYP9K1* diagnostics tools detect and track the spread of pyrethroid metabolic resistance

The development of the AS-PCR and LNA TaqMan assays to detect the 454A-*CYP9K1* mutation represents a significant advancement in the detection and monitoring of metabolic resistance in the field. These new diagnostic tools correlate strongly with pyrethroid resistance in both laboratory hybrid and field-collected mosquitoes, highlighting the significant role of allelic variation in *CYP9K1*-driven resistance. The *CYP9K1* assays will complement existing P450-based assays, such as *CYP6P9a* and *CYP6P9b*, currently used to track resistance in *An. funestus* populations in Southern Africa (Mugenzi *et al.* 2019; Weedall *et al.* 2019). Additionally, DNA-based diagnostic tools have previously been developed for other resistance markers, e.g. the 6.5-kb structural variant insertion between *CYP6P9a/b* genes shown to contribute to resistance in Southern Africa (Mugenzi *et al.* 2020), the recently detected 4.3-kb transposon-based marker in East and Central African *An. funestus* populations (Mugenzi *et al.* 2024) and the L119F-*GSTe2* marker, which confers pyrethroid/DDT resistance, used to track resistance in West and Central Africa (Riveron, Cristina, *et al.* 2014; Riveron, Ibrahim, *et al.* 2014).

## The mutant-type 454A-*CYP9K1* allele reduces the efficacy of pyrethroid nets

PCR genotyping of the G454A-*CYP9K1* marker in alive and dead samples has linked the 454A mutation to reduced insecticide efficacy. This highlights the growing challenge posed by metabolic resistance to malaria vector control, contributing to the recent increase in malaria incidence and mortality in Africa (WHO 2023). Previous studies have also associated the reduced efficacy of pyrethroid-treated nets with metabolic resistance markers *CYP6P9a*, the 6.5-kb structural variant insertion and *CYP6P9b* (Mugenzi *et al.* 2019, 2020; Weedall *et al.* 2019; Menze, Mugenzi, *et al.* 2022). However, nets treated with the synergist PBO were shown to be significantly more effective than conventional pyrethroid-only nets in killing mosquitoes in populations carrying the 454A-*CYP9K1* mutant allele. Therefore, the use of PBO or dual active ingredient bed nets should be prioritized in the regions where this mutation is prevalent.

## Africa-wide distribution pattern of 454A-*CYP9K1* mutation supports the existence of barriers to gene flow in *An. funestus* across the continent

Spatiotemporal analysis of the 454A-*CYP9K1* mutation revealed an increasing frequency over time in East Africa (Tanzania and Uganda) and Central Africa (Cameroon). This could be linked to increased selection pressure exerted by insecticide-based interventions and underscores the need for robust surveillance of insecticide resistance, especially for newly introduced insecticides like chlorfenapyr. The near fixation of the 454A-*CYP9K1* allele in Cameroon (Mibellon) suggests its significant role in conferring resistance in these localities. In contrast, no directional selection of the 454A-*CYP9K1* allele was observed in Southern Africa (Malawi and Mozambique) and West Africa (Benin and Ghana), indicating its limited role in these regions. Resistance in these regions is

primarily driven by other genes, notably the duplicated *CYP6P9a/b* (Ibrahim *et al.* 2015; Mugenzi *et al.* 2019; Weedall *et al.* 2019; Weedall *et al.* 2020; Mugenzi *et al.* 2023). The striking regional contrast of metabolic resistance markers in African populations of *An. funestus* populations highlights the challenges that barriers to gene flow pose to future interventions like gene drive or sterile insect techniques (SIT), which rely on the migration of genetically modified flies.

## Conclusion and future endeavors

This study has demonstrated that allelic variation of the *CYP9K1* gene is a major mechanism driving pyrethroid resistance in East and Central Africa. The resistance 454A-*CYP9K1* allele is a key driver of pyrethroid resistance, as illustrated by its near-fixation selection in sampled sites in East and Central African regions. This 454A-*CYP9K1* allele was shown to reduce the efficacy of pyrethroid-treated LLINs, but PBO-based nets were shown to have high efficacy even with the samples harboring the mutant-type allele. The DNA-based molecular diagnostic assay designed in this study is a robust tool for detecting and tracking the 454A-*CYP9K1* resistance marker in the field and assessing its impact on the efficacy of vector control tools.

Further studies could evaluate the correlation between this 454A-*CYP9K1* marker and resistance to other insecticide classes and its impact on newly recommended interventions, including dual active ingredient nets. Additionally, with the discovery of co-occurring new markers such as the 4.3-kb structural variant in *An. funestus* (Mugenzi *et al.* 2024), the combined effect of multiple markers could also be prioritized in future studies, along with the functional validation of other mutations detected during this study. Moreover, continuous surveillance of the 454A-*CYP9K1* marker should be prioritized across Africa using fresh samples, especially in regions shown to have moderate or low frequencies of the resistant allele.

## Data availability

The resource origin and associated information are described in the key resource table. The *CYP9K1* sequences for 2014 and 2020 *An. funestus* samples generated during this study have been deposited in the GenBank database (accession numbers: PP701030-PP701109 and PP701111-PP701270, respectively, for 2014 and 2020 *CYP9K1* sequences).

Supplemental material available at GENETICS online.

## Acknowledgments

We are grateful to the collection site communities for allowing us access to their sites for sampling. Special thanks to Helen Irving (LSTM) for reagents ordering and handling the sequencing, and to Dr. Mark Paine (LSTM) for providing the technical platform used for in vitro metabolism assays. The CYP9K1 reference sequence used for primer design and sequence analysis was downloaded from vectorbase database (www.vectorbase.org) using the accession number AFUN007549.

## Funding

This work was supported by a Wellcome Trust Senior Research Fellowships in Biomedical Sciences (217188/Z/19/Z) to CSW and a Bill and Melinda Gates Foundation Investment (INV-006003) to CSW. The funders had no role in study design, data collection and analysis, decision to publish, or preparation of the manuscript.

## Conflicts of interest

The author(s) declare no conflict of interest.

## Author contributions

Design and conceptualization of the study: C.S.W. Methodology: C.S.D.T., M.F.M.K., M.T., A.M., L.M.J.M., M.J.W., J.H., S.S.I., and C.S.W. Sample collection: C.S.D.T., M.T., R.F.T., M.G., and C.S.W. Investigation: C.S.D.T., M.F.M.K., A.M., and N.M.T.T. Visualization: C.S.D.T., M.T., J.H., S.S.I., and C.S.W. Supervision: M.T., L.M.J.M., S.S.I., M.H.D., and C.S.W. Writing—Original draft: C.S.D.T. and C.S.W. Writing—review and editing: C.S.D.T., M.F.M.K., M.T., A.M., L.M.J.M., J.H., S.S.I., M.H.D., and C.S.W.

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

*Editor: M. Lawniczak*