## [Peer Review File · Genetics]

A single mutation G454A in the P450 CYP9K1 drives pyrethroid resistance in the major malaria vector *Anopheles funestus* reducing bed net efficacy

Carlos Djoko Tagne, Mersimine Kouamo, Magellan Tchouakui, Abdullahi Muhammad, Leon Mugenzi, Nelly Tatchou-Nebangwa, Riccardo Thiomela, Mahamat Gadji, Murielle Wondji, Jack Hearn, Mbouobda Desire, Sulaiman S Ibrahim, and Charles Wondji

NOTE: The reviews and decision letters are unedited and appear as submitted by the reviewers.

In extremely rare instances and as determined by a Senior Editor or the EIC, portions of a review may be redacted. If a review is signed, the reviewer has agreed to no longer remain anonymous.

The review history appears in chronological order.

Review Timeline:

Submission Date:	2024-05-06
Editorial Decision:	2024-06-28
Resubmission Received:	2024-08-01
Editorial Decision:	2024-09-16
Resubmission Received:	2024-10-15
Accepted:	2024-10-24

June 28, 2024

GENETICS-2024-307066

A single mutation G454A in P450 CYP9K1 drives pyrethroid resistance in the major malaria vector *Anopheles funestus* reducing bed net efficacy

Dear Dr. Djoko Tagne:

Thank you for your submission to GENETICS. Your manuscript is not currently acceptable for publication in GENETICS, however, upon major revision addressing the majority of the comments of the reviewers, we would consider a revised manuscript.

All reviewers and myself agree the science is sound and the results are interesting.

However, you need to improve overall readability and narrative, and also ensure sufficiently clear background for a non-malaria/insecticide expert. Reviewer 1 took a lot of time to offer detailed feedback and edits, please do make all of those corrections with one exception: You do not need to split the manuscript into two papers but you do need to make the full story much easier to follow with revision to the writing to improve clarity. Please ensure you share a tracked changes version upon resubmission to enable revisions to be easily examined.

We look forward to receiving your revised manuscript. Please let the editorial office know approximately how long you expect to need for revisions.

Upon resubmission, please include:

1. A clean version of your manuscript;
2. A marked version of your manuscript in which you highlight significant revisions carried out in response to the major points raised by the editor/reviewers (track changes is acceptable if preferred);
3. A detailed response to the editor's/reviewers' feedback and to the concerns listed above. Please reference line numbers in this response to aid the editor and reviewers.

Your paper will likely be sent back out for review.

Additionally, please ensure that your resubmission is formatted for GENETICS
<https://academic.oup.com/genetics/pages/general-instructions>

Follow this link to submit the revised manuscript: Link Not Available

Sincerely,

Mara Lawniczak
Associate Editor
GENETICS

Approved by:
David Begun
Senior Editor
GENETICS

Reviewer #1 (Comments for the Authors (Required)):

Article Title: A single mutation G454A in P450 CYP9k1 drives pyrethroid resistance in the major malaria vector *Anopheles funestus* reducing bed net efficacy

Personal Message to Authors:

I am really impressed by the work in this paper and love the field-to-laboratory-to-field journey this endeavor takes us on as readers. While I offered a lot of information in terms of both major and minor edits, I hope you will interpret that as my desire to contribute to making this publication the best that it can be and helping you see this paper through the publication process. Congratulations on work well done.

Reviewer Summary:

The authors take on an ambitious suite of aims in this study in an effort to first track the global regional distribution of a mutant allele - found in a vector fly population capable of carrying and spreading malaria - over time, then demonstrate the mutant allele's resistance to insecticide and test for a positive correlation or relationship between the expression of this allelic variant and resistance to widely used insecticides in Africa, design a simple diagnostic tool to detect the resistance mutation in *An. funestus*, apply the diagnostic to track the efficacy of insecticide-treated bed nets in Africa, and, finally, offer an evaluation of what the presence, location, and fixation of this mutant allele means in the context of gene flow, microhabitats, and effective preventions for malaria across Africa. I have reduced all the components of the article into four main Aims: 1) Investigated genetic variation of the CYP9K1 gene in *An. funestus* across Africa and evaluated changes in allele frequencies over time, 2) through in vitro and in vivo experimental work, confirmed the relationship between mutant CYP9K1 and insecticide resistance and how that is impacted by gene expression, respectively, 3) designed a diagnostic assay around the G454A mutation (mutant allele) and ran field studies on the efficacy of insecticide-treated bed nets, and 4) discussed what the increased resistance to insecticide means for *An. funestus* population genetics given gene flow between fly populations and connectivity of habitats. Rarely do we see a field-to-laboratory-to-field study contained in a single publication. As someone who appreciates full-circle, whole story science, I would like to commend the authors on bringing so many different elements of the story under the umbrella of one article.

That being said, I would be remiss if I didn't suggest that the article could be more than one article, with Aims 1 and 2 comprising one strong article, and allowing more review of the different mutations explored from other work (researchers and publications) that have led to insecticide resistance in these flies and may serve as a natural experiment and/or offer predictive power of the fate of this mutation or how it may repattern the population genetics of this species based on strong directional selection and differential survival/reproduction of flies. A second paper could represent the design of the diagnostic, the field data of fly differential survival associated with efficacy of insecticide treatment of bed nets, and the discussion of what this means for fly population genetics over time (Aims 3 & 4), as well as human and animal health. I would like to reiterate that I like the story together, but that two papers are a possibility.

Should you choose to keep the whole story together in one publication, there are several changes that need to be made to increase the effectiveness of the publication and impact of the story that you are telling with this data. I would consider these major edits:

Major Edits:

1. To improve the readability of your publication and help your reader track the different aims, tests, and results, you could number your aims/goals as I did in the summary (doesn't have to be the same as what I did) and follow that throughout the paper, including introducing your aims as numbered at the end of your introduction, using the same numbering scheme to discuss the methods (numbered heading in bold, methods contained within Aim in smaller headings in italics under each Aim) and results. The structure of the discussion already supports this. This is the most important of the major edits. A strong and consistent re-organization of this paper by AIM is going to help your reader immensely and also allow you to keep track of the grouped tests and results.
2. I would also add a sentence to the end of your introduction selling this study as the field-to-laboratory-to-field study that it is. It is rare and should be highlighted and celebrated. See similar note for first paragraph of Discussion below.
3. Your abstract and introduction currently read well. Your methods section is where you start to lose your reader a bit, only because there is a lot of different things (Aims) going on here. The numbering scheme will help, but your method section is much too long (1/3 of your paper at the moment. You need to reduce the content here without losing reproducibility.
4. Your methods and results should be revisited and edited to more powerful explanations of tests/statistics. There were whole sections where it wasn't clear from the writing that a statistical test had been done (see lines 543-564 for example), and parts where the explanation of the results were unclear (see minor edits for line items). Restructuring the document by Aims, as suggested in 1, will help with this too, because it will ensure that each Aim/question/hypothesis is accompanied by a definitive test and result that the reader can track. One specific note is that I don't remember seeing use of Fisher's exact test mentioned in methods, but used in whole sections in results. Again, keeping grouped methods and results by Aims will help you not overlook these details.
5. While I normally reserve line item edits for minor edits, there were a high number of grammar and punctuation errors throughout the document that need to be corrected. There are also areas of duplicated information (see lines 186-197 for example), areas where I feel you are losing impact due to readability and incompleteness (see lines 119-132 and 858-868 for example and lines 997-1000), and several places where I feel you could make more of an impact by telling your reader how the details you are listing contribute to a bigger picture (see 7 for Discussion notes on niche content). You also should do a global search for "Nevertheless," "In the contrary," "very", "Statistically," and remove this language. These are unnecessary and just

taking up word space.

6. There are significant parts of the paper where the language of the paper reads niche to the malaria/malaria vector genetics community of scientists. Work done to soften technical language and acronyms that may be a bit more niche and less known by a wider genetics community, as well as more detailed information about the history of the work on resistance genes and how this work has translated to better understanding of vector genetics or application of knowledge gained to preventative or translational medicine and global health, will increase your reach and impact on your audience.

7. Figures/Tables:

- a. I think you could make your Figures have more impact if the Figure title was a statement of the main conclusion from the group of representations. Especially, since each figure represents an Aim in the study. This ties back in with the suggestion of numbering all these Aims throughout the paper to help your reader track tests/results.
- b. Figure 1A/B - I think these are fine the way they are, but might be more effective if each country/lab colony had it's own trajectory line and the x-axis was just the two time points. That way you can clearly see the changes over time (increase/decrease) in each line (country or lab colony). I am not sure that you can infer strong directional selection from Figures A/B alone as your Figure caption suggests, without showing pattern deviates from rest of gene/genome. I would remove that part from the A/B description line and save for discussion where you can discuss with other data that helps you reach that conclusion. Figures C/D - figure is fine, but you should briefly tell your reader what coding length tells you/why used (another measure of diversity or way to look at allelic variation?). E/F Ok. You might consider moving G/H to supplement to save space, as well as showing it along with the same table for 2014. You could then reference the 2014/2020 amino acid table in Figure 1 and in text along with Figure 1 as evidence for directional selection.
- c. Figure 2 is compelling. I am not sure if A/B are necessary for Figure 2 or could be relegated to supplement. You might want to justify its necessity/remind reader what it means and why it was important to show in main document.
- d. Figure 3 is satisfying. My only note is that a white instead of blue background in panel B may be more ideal for visualizing.
- e. Figure 4 and 5 also good.
- f. Table S4 is missing a decimal point in the reporting associated with Elende F1 deltamethrin RR vs RS (I think).

8. The Discussion remains the most niche of all the content. You should review the discussion and add in content that puts your results in broader context where you can. Your first paragraph of the discussion should be rewritten to increase the impact and be more complete in the dissection of all the important results, listed by Aim. You should re-emphasize that this was a field-to-laboratory-to-field study and why that is important, rare and more impactful than studies that might address this system in the field or in the laboratory only. You should justify your determination of directional selection, even if it seems obvious to you, by summarizing all your results that support that conclusion, and eliminates alternative explanations. You need to add a heading to your last paragraph "Conclusions and Future Studies" and offer your reader some ideas of how this paper and it's results should be pushed forward for future work, whether that be applications in the field, additional tests needed in laboratory, future field monitoring, or something else. You should also consider limitations and offer your reader a few sentences/thoughts in this final paragraph of things that may have limited your work that could be improved in future studies.

Minor Edits:

40/41 "malaria vectors."

154 remove close parentheses or add open

182 "used" ????

286 "wild-type (FANG) allele for the gene"

287 remove ", "

292 remove "proceeded"

306 remove "construct"

312 remove "papers"

316/317 remove parentheses

323/324 rewrite sentence to be more clear and concise

327 remove "(primers" and end parentheses

353 remove space before "parameters". Is what follows a list of the modifications? If so, used colon and list. If not, briefly list modifications.

369/371 long and confusing sentence

393 ", and Permanet 3.0 side. . ."

451 remove "2014 samples"

454 remove "very"

469 "independently from Uganda and Cameroon and from each other (Figure 1D)."

470-474 rewrite to make more concise

447-487 condense so you are not repeating the same info for diversity measures and haplotypes

489/490 ". . .(40/40) and at low frequency in Cameroon (9/40) and Malawi (13/40) in 2014. . ."

488 "alanine"

524 substitute "higher" for "significantly greater"

525 lower case "p" for p-value throughout

535 "wild-type"
536-540 combine the direct comparisons (i.e., 18.26% vs 37.27%, 16.83% vs. 32.77%, etc) for the two genotypes
543-564 reduce bioassays to a couple of lines at end of preceding paragraph and reference a table and figure.
599 move figure reference to 595
640 should that be (R) instead of (S)
644 "exposure and 0.05%. . ."
649 remove "In the Contrary,"
650 move 72% frequency inside parentheses
652 remove "Statistically,"
658 "A similar. . ."
667 "samples, while 67%"
704-709 rewrite to make more clear and concise
712 remove "oppositely,"
710-714 rewrite to make more clear and concise
710-730 reduce to 4 sentences or less
738/785 remove "Nevertheless,"
770-856 Somewhere in here please report the number of flies sampled (N=?)
788-791 rewrite to make more clear and concise
796-797 put percentages in parentheses
866 "track" instead of "tract"
893 remove duplicate "and the"
952 "contributes"
963 Maybe start this paragraph with sentence that begins on line 965
969 "contributing to the. . ."
982 remove "mostly"
984 "highlighting the need"
985 "insecticide resistance"
986 "are being. . ."
990 "evidence" / remove "contrastingly"
994 remove "very"
995 "genes,"
997-1000 unclear, please elaborate
1001 add "Conclusions and Future Endeavors" heading
1004 "across East and Central Africa."

Reviewer #2 (Comments for the Authors (Required)):

The paper comprehensively investigates the notion that the mutation G454A in the CYP9K1 gene in the mosquito, *Anopheles funestus*, confers resistance to different classes of pyrethroids. The authors included a longitudinal study of CYP9K1 haplotypes in East and Central Africa, indicating selection for a predominant haplotype first found in Uganda in 2014, which contains the G454A mutation. Additionally, *in vitro*, and *in vivo* assays were performed to assess the performance of mutant and wild-type alleles of CYP9K1. All results support the hypothesis that the G454A mutation in the CYP9K1 gene found in *Anopheles funestus* is a major driver conferring pyrethroid resistance. The paper is extensive, sometimes repetitive, and very long. It was not an easy paper to read. I would consider editing to shorten it. However, the methods applied were appropriate to the problem, the results very convincing and the interpretation sound.

MAJOR CONCERNS

The quality of the writing is uneven throughout the paper. Specifically, the prose in the Introduction section is poor. For example, lines 63-64 "...significant progress made since the years 2000s" and lines 72-73 "...These control tools were attributed more than 70% of the decrease in malaria mortality...". Suggest editing to improve.

Anopheles funestus is a species complex including 13 sibling species. Authors should confirm they are working with *Anopheles funestus sensu stricto*.

The authors should provide sample sizes throughout.

Authors should provide a justification for the *Drosophila* work.

Authors should be consistent in the terms used to refer to the mutation. The authors use too many variations, such as G454A-CYP9K1, CYP9K1-G454A, (G454A) of CYP9K1, 454ACYP9K1, 454A, and 454A1a. This contributes to the difficulty in reading this long paper.

The *kdr* mutation has recently been identified in *A. funestus*. This mutation also confers

resistance to pyrethroid insecticides and it has been shown that there is an interaction between this mutations and mutations in Cyp9K1, What was the kdr genotype for the A. funestus strains/populations used in this study?

MINOR CONCERNS

75-76 This statement requires further explanation to clarify

200 "About 4 replicates..." Be specific.

Figure 1: Remove an "A" letter from graph A if it doesn't mean anything.

Figure 2: Add titles with the name of the insecticide type to graphs E, F, and G.

Figure 5: Clarify what the Cameroon pie chart in graph A represents and if there is any relation to graph B.

Reviewer #3 (Comments for the Authors (Required)):

The authors looked at the mutation G454A-CYP9K1 in *An. funestus*, a mutation that had been previously identified as being fixed in a population resistant to pyrethroids in Uganda. Their analysis is quite thorough as they observed the effect of the mutation when artificially boosted in *E. coli* and *Drosophila melanogaster*, compared the proportion of R and S alleles in alive and dead *An. funestus* using several experimental methods (WHO tubes, cone assays, EHTs) and several different pyrethroids (Type I, Type II and Type II + PBO), looked at the changes in distribution of the mutation across Africa and created 2 PCR-based assays to quickly detect the mutation in wild populations. Overall, their conclusions seem sound and I would say that the quality of their work is commendable. However, in my opinion, this manuscript has several weaknesses that need to be addressed.

Major recommendation:

The manuscript covers a lot of ground but not always in a very clear way. For instance, the creation of the PCR-based assays is a major contribution but it barely figures in the Results section. It is mentioned in later section but it mostly features in the Methods section leaving the reader with the impression that it already existed and is mainly used as a tool. In a very different way, the authors mention that "PBO bed nets should be prioritized in the regions where that mutation is present." but that has little to do with the prevalence of 454A-CYP9K1 and more to do with the fact that mortality in the presence of PBO is still 100%, a fact that is mentioned almost in passing.

A significant part of the results section is about the different assays using different methods and different insecticides. In all (but one, I think) cases, the result goes: RR and RS are more prevalent among alive than dead, while SS is the other way round. The authors try to find creative ways to say this repeatedly but it ends up being more confusing and wordy than needed. I could have said in the previous point that a figure is better than many words. The authors do provide figures but they are all very similar and (more importantly) untitled, leaving the readers to have to dig through the caption of the aggregated figure to discover which one corresponds to which insecticide.

In the Discussion section, one of the key findings is that the frequency of the R allele is at high frequency or fixed in Central ("in Cameroon [...] the frequency of the 454A-CYP9K1 moved from 28.5% in 2014 to 94% in 2021") Africa while "no evidence of directional selection of this mutant-type 454A-CYP9K1 allele was observed in Southern Africa (Malawi and Mozambique)". This is simply incorrect. RR is almost fixed in one location in Cameroon (Mibellon) but at low to intermediate frequencies in the rest of the country indicating that the geographical distribution of the RR genotype might be more complex than that. Furthermore, most results show that the heterozygous genotype might confer enough resistance and the distribution in Southern Africa in 2020 or 2021 looks a lot like the distribution in Elende or Tibati in 2021. The same section also considers that it is possible that the resistance haplotype spread from Uganda to Cameroon. That seems possible but one would have to explain why this spread is only visible in 1 site in Cameroon and not the closest to Uganda.

In some cases, the authors make their points in a confusing order. For instance, the bioassays are presented in the order alphacypermethrin, deltamethrin, permethrin; while the tube bioassays are permethrin, alphacypermethrin (for the crosses); and alphacypermethrin, permethrin, deltamethrin (for the wild-caught). Similarly, the discussion goes from "Allelic variation in CYP9K1 combines with its over-transcription to confer higher pyrethroid resistance intensity" (very much insecticide resistance focused) to "Novel G454A-CYP9K1 diagnostics are reliable DNA-based tools to detect and track the spread of metabolic resistance to pyrethroid insecticides" (about the importance of the assays) and back to "Mutant-type 454A-CYP9K1 allele reduces the efficacy of pyrethroid nets" (once again about insecticide resistance).

Minor comments:

There are many situations where the English is either a little awkward (e.g., "predominant [...] than" l.623-624) or where the wrong word is used (e.g. "we unwound the role" l.861 is probably "we uncovered the role"). I think this sometimes stems from the authors avoiding to use the same phrase structure to make the manuscript easier to read but this leads to "Similarly" (l.744) being used to connect two sentences that literally show opposite behaviours and "Interestingly" (l.748) being used to observe that RS has an increased survivorship compared to SS, which has also been true for every other assay. The worst offender is probably "identified" at line 487 that can lead a reader to assume that this is the first time that anyone observed G454A-CYP9K1. I have some issues with Figure 1 and the associated text:

In Figure 1A and B, only the dots representing the same country should be connected. As it is, the reader is left assuming that the relationship between, say Cameroon 2014 and Cameroon 2020 is similar to the one between Malawi 2020 and Ghana 2020. I would agree that Figure 1A and B show a "signature of strong directional selection" for Cameroon but it is less obvious for

Uganda, where haplotype and nucleotide diversity were already quite low.

The authors do not mention, in relation to Figure 1G, that 454A-CYP9K1 is also present is also fixed for FUMOZ.

The authors highlight the mutation G979A-CYP9K1 (exclusive to Ghana) for no clear reason.

Some other mutation (say A735-CYP9K1 or G738C-CYP9K1) are fixed in Uganda, Cameroon and FUMOZ, present in Malawi but absent from FANG and Ghana, a very similar pattern to G454A-CYP9K1. They should be mentioned (in particular in light of the mention of G979A-CYP9K1) if only to say that they are not the subject of this manuscript.

Figure 1E and F appear after G and H.

I am not sure all readers will be very familiar with the various types of bednets and I think it would be helpful to remind, at the beginning of the section on EHTs and on the plots for the various bednets, which insecticide they use to better connect that section to the rest of the assays.

Times are sometimes shown as "hours" (l. 318, by the way, it should be "2 hours" and not "2hours"), "h" (l. 545) or "hrs" (l. 553).

The authors should pick one and stick to it.

Some references (e.g., "WHO, 2022", l. 72) use a comma, others don't (e.g., "WHO 2022", l.76).

A reference is needed l. 527.

The second sentence of "Mosquito samples collection and rearing" is barely readable. More than 1 sentence can be used and it might be worth pointing to the maps (Figures 5A and B) or create a new one.

For some reason, lines 329 and 330 are empty.

On line 601, I think "Figure 4A and B" is supposed to be "Figure 3A and B" as it is the reference at the end of the sentence.

Furthermore, I don't think they are both necessary.

The last 2 sentences of "Allele specific PCR (AS-PCR) design for G454A-CYP9K1 mutation" can probably be combined as they are very similar.

On line 613, "In Contrary" is supposed to be "Contrarily", "However", "On the other hand", or maybe "On the contrary".

On line 640, I assume that "454A-CYP9K1 mutant type allele (S)" is supposed to be "454A-CYP9K1 mutant type allele (R)".

On line 717, "the CYP9K1" is probably supposed to be "the CYP9K1 mutation" or "454A-CYP9K1".

Yaoundé, 1st August 2024

Dear Editor, *GENETICS* Journal

Ref: Research paper revision (Manuscript code Ref: GENETICS-2024-307066)

We are grateful to the editors/reviewers for their comments on the current manuscript titled "**A single mutation G454A in P450 CYP9K1 drives pyrethroid resistance in the major malaria vector *Anopheles funestus* reducing bed net efficacy**", which greatly improved the quality of this revised manuscript. We have revised the entire manuscript according to the editors'/reviewers' comments and the list of changes made is presented below and also highlighted in the "track changes" version submitted.

Kindly find below the point-by-point responses to the comments.

Yours Sincerely,

Correspondence to:

Charles Sinclair Wondji
Liverpool School of Tropical Medicine (LSTM)
Pembroke Place, Liverpool L3 5QA, United Kingdom
Tel: +44 (0)151 705 3140/ +237 653 157 384
E-mail: charles.wondji@lstmed.ac.uk

Carlos Simeon Djoko Tagne
Centre for Research in Infectious Diseases (CRID)/ University of Bamenda
Nkoulou-Carriere, Yaounde, Cameroon
Tel: +237 651 060 395
E-mail: carlos.djoko@crid-cam.net

Reviewer #1 (Comments for the Authors (Required)):

Article Title: A single mutation G454A in P450 CYP9k1 drives pyrethroid resistance in the maj or malaria vector *Anopheles funestus* reducing bed net efficacy

Personal Message to Authors:

I am really impressed by the work in this paper and love the field-to-laboratory-to-field journey this endeavor takes us on as readers. While I offered a lot of information in terms of both major and minor edits, I hope you will interpret that as my desire to contribute to making this publication the best that it can be and helping you see this paper through the publication process. Congratulations on work well done.

Authors appreciate this kind comment from Reviewer#1. We appreciate the time and efforts the Reviewer invested for the feedback which improved the quality of this manuscript.

Reviewer Summary:

The authors take on an ambitious suite of aims in this study in an effort to first track the global regional distribution of a mutant allele

- found in a vector fly population capable of carrying and spreading malaria

- over time, then demonstrate the mutant allele's resistance to insecticide and test for a positive correlation or relationship between the expression of this allelic variant and resistance to widely used insecticides in Africa, design a simple diagnostic tool to detect the resistance mutation in *An. funestus*, apply the diagnostic to track the efficacy of insecticide-

treated bed nets in Africa, and, finally, offer an evaluation of what the presence, location, and fixation of this mutant allele means in the context of gene flow, microhabitats, and effective preventions for malaria across Africa. I have reduced all the components of the article into four main Aims: 1) Investigated genetic variation of the CYP9K1 gene in *An. funestus* across Africa and evaluated changes in allele frequencies over time, 2) through in vitro and in vivo experimental work, confirmed the relationship between mutant CYP9K1 and insecticide resistance and how that is impacted by gene expression, respectively, 3) designed a diagnostic assay around the G454A mutation

(mutant allele) and ran field studies on the efficacy of insecticide-treated bed nets, and 4) discussed what the increased resistance to insecticide means for *An. funestus* population genetics given gene flow between fly populations and connectivity of habitats. Rarely do we see a field-to-laboratory-to-

field study contained in a single publication. As someone who appreciates full-

circle, whole story science, I would like to commend the authors on bringing so many different elements of the story under the umbrella of one article.

That being said, I would be remiss if I didn't suggest that the article could be more than one article, with Aims 1 and 2 comprising one strong article, and allowing more review of the different mutations explored from other work

(researchers and publications) that have led to insecticide resistance in these flies and may serve as a natural experiment and/or offer predictive power of the fate of this mutation or how it may repattern the population genetics of this species based on strong directional selection and differential survival/reproduction of flies. A second paper could represent the design of the diagnostic, the field data of fly differential survival associated with efficacy of insecticide treatment of bed nets, and the discussion of what this means for fly population genetics over time (Aims 3

& 4), as well as human and animal health. I would like to reiterate that I like the story together, but that two papers are a possibility.

Should you choose to keep the whole story together in one publication, there are several changes that need to be made to increase the effectiveness of the publication and impact of the story that you are telling with this data. I would consider these major edits:

Major Edits:

1. To improve the readability of your publication and help your reader track the different aims, tests, and results, you could number your aims/goals as I did in the summary (doesn't have to be the same as what I did) and follow that throughout the paper, including introducing your aims as numbered at the end of your introduction, using the same numbering scheme to discuss the methods (numbered heading in bold, methods contained within Aim in smaller headings in italics under each Aim) and results. The structure of the discussion already supports this. This is the most important of the major edits. A strong and consistent re-organization of this paper by AIM is going to help your reader immensely and also allow you to keep track of the grouped tests and results.

Thanks to the reviewer for the suggestion. The manuscript has been reorganised accordingly as can be seen in the track-change version, for example in Lines 145-147 for the Methodology, Lines 327-328, Lines 371-373, 386-388 for the results.

2. I would also add a sentence to the end of your introduction selling this study as the field-to-laboratory-to-

field study that it is. It is rare and should be highlighted and celebrated. See similar note for first paragraph of Discussion below.

As suggested by the reviewer, a sentence has been added at the end of the introduction (Lines 129-142) and discussion section (Lines 631-642), to highlight that this study as the field-to-laboratory-to-field study.

3. Your abstract and introduction currently read well. Your methods section is where you start to lose your reader a bit, only because there is a lot of different things (Aims) going on here. The numbering scheme will help, but your method section is much too long (1/3 of your paper at the moment. You need to reduce the content here without losing reproducibility.

The methodology has been reduced accordingly and numbered accordingly as suggested, for example Lines 145-147 with some details transferred to the Supplementary text, for example Line 176.

4. Your methods and results should be revisited and edited to more powerful explanations of tests/statistics. There were whole sections where it wasn't clear from the writing that a statistical test had been done (see lines 543-564 for example), and parts where the explanation of the results were unclear (see minor edits for line items). Restructuring the document by Aims, as suggested in 1, will help with this too, because it will ensure that each Aim/question/hypothesis is accompanied by a definitive test and result that the reader can track. One specific note is that I don't remember seeing use of Fisher's exact test mentioned in methods, but used in whole sections in results. Again, keeping grouped methods and results by Aims will help you not overlook these details.

This has been done accordingly. The methods and results have been extensively revised, well edited and numbered per aim as suggested. Fisher's exact test mentioned in the methods section, under the Statistical Analysis (lines 321-323) was used to establish the correlations between G454A-CYP9K1 genotypes (RR, RS, SS) and resistance phenotype (Alive and dead).

5. While I normally reserve line item edits for minor edits, there were a high number of grammar and punctuation errors throughout the document that need to be corrected. There are also areas of duplicated information (see lines 186-197 for example), areas where I feel you are losing impact due to readability and incompleteness (see lines 119-132 and 858-868 for example and lines 997-1000), and several places where I feel you could make more of an impact by telling your reader how the details you are listing contribute to a bigger picture (see 7 for Discussion notes on niche content). You also should do a global search for "Nevertheless," "In the contrary," "very",

"Statistically," and remove this language. These are unnecessary and just taking up word space.

We agree with the reviewer comments and corrections have been made accordingly. Please see for example the duplicated information has been removed lines 1164-165. The introduction has been updated to make more impact Lines 129-142, lines 628-642, and Line 766-771. Global search has been made to remove "Nevertheless,", "In the contrary,", "very", "Statistically," for example Lines 459, 462, 469,471,479,482,490 and 497.

6. There are significant parts of the paper where the language of the paper reads niche to the malaria/malaria vector genetics community of scientists. Work done to soften technical language and acronyms that may be a bit more niche and less known by a wider genetics community, as well as more detailed information about the history of the work on resistance genes and how this work has translated to better understanding of vector genetics or application of knowledge gained to preventative or translational medicine and global health, will increase your reach and impact on your audience.

We highly appreciate this comment from the reviewer. In the revised version, we tried to soften as much as possible the technical language and acronyms to make it more generalizable to non-experts.

7. Figures/Tables:

a. I think you could make your Figures have more impact if the Figure title was a statement of the main conclusion from the group of representations. Especially, since each figure represents an Aim in the study. This ties back in with the suggestion of numbering all these Aims throughout the paper to help your reader track tests/results.

This has been updated accordingly and numbered per aim, thanks to the reviewer for the remark. Please check lines 362, 414, 500, 561-562.

b. Figure 1A/B

- I think these are fine the way they are, but might be more effective if each country/lab colony had it's own trajectory line and the x-axis was just the two time points. That way you can clearly see the changes over time (increase/decrease) in each line (country or lab colony). I am not sure that you can infer strong directional selection from Figures A/B alone as your Figure caption suggests, without showing pattern deviates from rest of gene/genome. I would remove that part from the A/B description line and save for discussion where you can discuss with other data that helps you reach that conclusion. Figures C/D - figure is fine, but you should briefly tell your reader what coding length tells you/why used

(another measure of diversity or way to look at allelic variation?). E/F Ok. You might consider moving G/H to supplement to save space, as well as showing it along with the same table for 2014. You could then reference the 2014/2020 amino acid table in Figure 1 and in text along with Figure 1 as evidence for directional selection.

This has been modified and updated accordingly, Figures A and B updated accordingly as suggested by the reviewer and figures G/H moved to the Supplement Figure S2 (lines 360-361, Figure 1). The coding length because changes in the coding sequence could directly affect protein structure and function,

c. Figure 2 is compelling. I am not sure if A/B are necessary for Figure 2 or could be relegated to supplement. You might want to justify its necessity/remind reader what it means and why it was important to show in main document.

Figure 2 has been updated accordingly as suggested (Lines 412-413). A/B have been relegated to Supplement Figure S3.

d. Figure 3 is satisfying. My only note is that a white instead of blue background in panel B may be more ideal for visualizing.

We agree with the reviewer that a white background in panel B could be more ideal for visualizing, unfortunately it is not possible to make any change as this is automatically generated by the AriaMX Real-Time qPCR software.

e. Figure 4 and 5 also good.

f. Table S4 is missing a decimal point in the reporting associated with Elende F1 deltamethrin RR vs RS (I think).

This has been corrected according to the reviewer comment.

8. The Discussion remains the most niche of all the content. You should review the discussion and add in content that puts your results in broader context where you can. Your first paragraph of the discussion should be rewritten to increase the impact and be more complete in the dissection of all the important results, listed by Aim. You should re-emphasize that this was a field-to-laboratory-to-field study and why that is important, rare and more impactful than studies that might address this system in the field or in the laboratory only. You should justify your determination of directional selection, even if it seems obvious to you, by summarizing all your results that support that conclusion, and eliminates alternative explanations.

The discussion has been reviewed accordingly as suggested, for example see Lines 629-642, 644-649.

You need to add a heading to your last paragraph "Conclusions and Future Studies" and offer your reader some ideas of how this paper and its results should be pushed forward for future work, whether that be applications in the field,

additional tests needed in laboratory, future field monitoring, or something else. You should also consider limitations and offer your reader a few sentences/thoughts in this final paragraph of things that may have limited your work that could be improved in future studies.

Thanks very much to the reviewer for these remarks. A heading “Conclusion and Future Endeavours” has been added as suggested by the reviewer (Lines 772-788).

Minor Edits: We agree with the editor/reviewer and are grateful for all these remarks, assuring the editor/reviewer that we have all addressed them accordingly.

40/41 "malaria vectors." Corrected (Line 41).

154 remove close parentheses or add open Corrected (Line 204).

182 "used" ??? Updated (Line 161-163).

286 "wild-type (FANG) allele for the gene" corrected (Line 198).

287 remove "," Corrected (Line 198).

292 remove "proceeded" Corrected (Line 203).

306 remove "construct" Corrected (Supplementary text Line 74).

312 remove "papers" Corrected (Line 215).

316/317 remove parentheses Corrected (Line 219).

323/324 rewrite sentence to be more clear and concise Corrected (Lines 224-225).

327 remove "(primers" and end parentheses Corrected (Line 228).

353 remove space before

"parameters". Is what follows a list of the modifications? If so, used colon and list. If not, briefly list modifications. Corrected (Line 242).

369/371 long and confusing sentence Corrected (Lines 246-254).

393 ", and Permanet 3.0 side. . ." Corrected (Line 267).

451 remove "2014 samples" Corrected (Line 329).

454 remove "very" Corrected (Line 330).

469 "independently from Uganda and Cameroon and from each other (Figure 1D)." Corrected (Line 339/340).

470-474 rewrite to make more concise Corrected (Lines 341-344).

447-

487 condense so you are not repeating the same info for diversity measures and haplotypes

489/490 ". . . (40/40) and at low frequency in Cameroon (9/40) and Malawi (13/40) in 2014. . ."

Corrected (Lines 329-350 and lines 351-357).

488 "alanine" Corrected (Line 352).

524 substitute "higher" for "significantly greater" Corrected (Line 383).

525 lower case "p" for p-value throughout Corrected throughout. For example, Line 383, 391, 392. Thanks to the reviewer.

535 "wild-type" Corrected (Line 393).

536-540 combine the direct comparisons (i.e., 18.26% vs 37.27%, 16.83% vs. 32.77%, etc) for the two genotypes Corrected (Line 391-392).

543-

564 reduce bioassays to a couple of lines at end of preceding paragraph and reference a table and figure. Corrected (Line 394-406).

599 move figure reference to 595 Corrected (Line 445, 447).

640 should that be (R) instead of (S) Corrected (Line 474).

644 "exposure and 0.05% . . ." Corrected (Line 477).

649 remove "In the Contrary," Corrected (Line 479).

650 move 72% frequency inside parentheses Corrected (Line 489).

652 remove "Statistically," Corrected (Line 482).

658 "A similar. . ." Corrected (Line 492).

667 "samples, while 67%" Updated (Line 492).

704-709 rewrite to make more clear and concise Corrected (Lines 522-528).

712 remove "oppositely," Corrected (Line 530).

710-714 rewrite to make more clear and concise Corrected (Lines 529-533).

710-730 reduce to 4 sentences or less Updated (Lines 529-535 and Lines 536-542).

738/785 remove "Nevertheless," Corrected (Line 551).

770-856 Somewhere in here please report the number of flies sampled (N=?) Reported throughout the manuscript for example lines 390/579/599/600.

788-791 rewrite to make more clear and concise Corrected (Line 585-587).

796-797 put percentages in parentheses Corrected (Line 585-587).

866 "track" instead of "tract" Corrected (Line 639).

893 remove duplicate "and the" Corrected (Line 667).

952 "contributes" Corrected (Line 724).

963 Maybe start this paragraph with sentence that begins on line 965 Corrected (Line 735).

969 "contributing to the. . ." Corrected (Line 738).

982 remove "mostly" Corrected (Line 752).

984 "highlighting the need" Corrected (Line 754).

985 "insecticide resistance" Corrected (Line 755).

986 "are being. . ." Corrected (Line 755).

990 "evidence" / remove "contrastingly" Corrected (Line 759/760).

994 remove "very" Corrected (Line 763).

995 "genes," Corrected (Line 764).

997-1000 unclear, please elaborate Corrected (Lines 766-771).

1001 add "Conclusions and Future Endeavors" heading Corrected (Line 772).

1004 "across East and Central Africa." Corrected (Line 776).

Reviewer #2 (Comments for the Authors (Required)):

The paper comprehensively investigates the notion that the mutation G454A in the CYP9K1 gene in the mosquito, *Anopheles funestus*, confers resistance to different classes of pyrethroids.

The authors included a longitudinal study of CYP9K1 haplotypes in East and Central Africa, indicating selection for a predominant haplotype first found in Uganda in 2014, which contains

the G454A mutation. Additionally, in vitro, and in vivo assays were performed to assess the performance of mutant and wild-

type alleles of CYP9K1. All results support the hypothesis that

the G454A mutation in the CYP9K1 gene found in *Anopheles funestus* is a major driver conferring pyrethroid resistance. The paper is extensive, sometimes repetitive, and very long. It

was not an easy paper to read. I would consider editing to shorten it. However, the methods applied were appropriate to the problem, the results very convincing and the interpretation sound.

Authors are grateful for Reviewer # 2's painstaking efforts and time to make this revised version of this paper better. The paper has been extensively revised and the content summarised without losing the originality of the findings as you can see in the track-change version.

MAJOR CONCERNS

The quality of the writing is uneven throughout the paper. Specifically, the prose in the

Introduction section is poor. For example, lines 63-64 "...significant progress made since the years 2000s" and lines 72-73 "...These control tools were attributed more than 70% of the decrease in malaria mortality...". Suggest editing to improve.

This has been corrected accordingly to improve the introduction (Lines 63-72) and the entire paper as well.

Anopheles funestus is a species complex including 13 sibling species. Authors should confirm they are working with *Anopheles funestus sensu stricto*.

Authors have confirmed that this work was carried out exclusively with *Anopheles funestus sensu stricto* (Line 162/163).

The authors should provide sample sizes throughout.

The sample size (n) has been provided throughout the revised version of the manuscript (Lines 390/579/599/600).

Authors should provide a justification for the *Drosophila* work.

Transgenic *Drosophila melanogaster* flies were used to investigate if overexpressing the *CYP9K1* gene alone could drive resistance to pyrethroid in these transgenic flies where other pyrethroid resistance genes are not expressed. Also, this was done to compare the resistance provided by the mutant-type to the wild-type alleles (Lines 196-198).

Authors should be consistent in the terms used to refer to the mutation. The authors use too many variations, such as G454A-CYP9K1, CYP9K1-G454A, (G454A) of CYP9K1, 454ACYP9K1, 454A, and 454Ala. This contributes to the difficulty in reading this long paper.

The variations have been corrected and homogenised accordingly. Please see lines 181, 240, 427.

The *kdr* mutation has recently been identified in *A. funestus*. This mutation also confers resistance to pyrethroid insecticides and it has been shown that there is an interaction between these mutations and mutations in *Cyp9K1*. What was the *kdr* genotype for the *A. funestus* strains/populations used in this study?

Thanks very much to the reviewer for pointing out this major discovery of *kdr* mutation in *An. funestus*. We have indicated in the revised version (Lines 98-100) this new paper (Odero et al. 2023; preprint "Discovery of knock-down resistance in the major African malaria vector

1 *Anopheles funestus*’). We agree with the reviewer that the possibility that the *kdr* and *CYP9K1* could combine to exacerbate pyrethroid resistance is not excluded, although the L976F *kdr* detected by the authors was not linked with pyrethroid resistance but rather with DDT resistance and only in one locality (Morogoro) in Tanzania. Nevertheless, as suggested by the reviewer, we have now highlighted the discovery of this *kdr* mutation in *An. funestus* in the introduction (Lines 98-100).

MINOR CONCERNS

75-76 This statement requires further explanation to clarify.

We agree with the reviewer and have provided further explanations accordingly (Lines 75-78).

200 "About 4 replicates..." Be specific. Corrected (Line 256).

Figure 1: Remove an "A" letter from graph A if it doesn't mean anything.

We would like to apologize for the mistake, it has been corrected (Figure 1, Lines 360-361).

Figure 2: Add titles with the name of the insecticide type to graphs E, F, and G.

Titles have been added with names of insecticides (Figure 2, Lines 412-413).

Figure 5: Clarify what the Cameroon pie chart in graph A represents and if there is any relation to graph B.

Thanks to the reviewer for pointing this out. Cameroon pie chart in graph A represents the genotype distribution in Mibellon district in 2021 (Single locality and single time point). This was related to Figure B, where we represented the genotype frequency in Mibellon at different time points (in 2014, 2018 and 2021) to demonstrate the pattern of evolution of the marker in Mibellon. Figure 5a have been updated by removing the Cameroon pie chart from the Africa-wide distribution graph as this was a duplicate. The Mibellon temporal genotype pie chart is presented on the Cameroon-wide distribution map alongside other localities (Figure 5a, Line 607-608).

Reviewer #3 (Comments for the Authors (Required)):

The authors looked at the mutation G454A-

CYP9K1 in *An. funestus*, a mutation that had been previously identified as being fixed in a po

pulation resistant to pyrethroids in Uganda. Their analysis is quite thorough as they observed the effect of the mutation when artificially boosted in *E. coli* and *Drosophila melanogaster*, compared the proportion of R and S alleles in alive and dead *An. funestus* using several experimental methods (WHO tubes, cone assays, EHTs) and several different pyrethroids (Type I, Type II and Type II + PBO), looked at the changes in distribution of the mutation across Africa and created 2 PCR-based assays to quickly detect the mutation in wild populations. Overall, their conclusions seem sound and I would say that the quality of their work is commendable. However, in my opinion, this manuscript has several weaknesses that need to be addressed.

Many thanks to the reviewer for the appreciations and most importantly for the time and all the efforts to go through this manuscript.

Major recommendation:

The manuscript covers a lot of ground but not always in a very clear way. For instance, the creation of the PCR-based assays is a major contribution but it barely figures in the Results section. It is mentioned in later section but it mostly features in the Methods section leaving the reader with the impression that it already existed and is mainly used as a tool.

We agree with the reviewer that in the initial submission we did not put major emphasis on the design of a PCR-based assay which is the major outcome of the study. This has been done now in the revised version, making it clearer that this is a major contribution in this paper (Lines 425-453).

In a very different way, the authors mention that "PBO bed nets should be prioritized in the regions where that mutation is present." but that has little to do with the prevalence of 454A-CYP9K1 and more to do with the fact that mortality in the presence of PBO is still 100%, a fact that is mentioned almost in passing. A significant part of the results section is about the different assays using different methods and different insecticides. In all (but one, I think) cases, the result goes: RR and RS are more prevalent among alive than dead, while SS is the other way round. The authors try to find creative ways to say this repeatedly but it ends up being more confusing and wordy than needed. I could have said in the previous point that a figure is better than many words. The authors do provide figures but they are all very similar and

(more importantly) untitled, leaving the readers to have to dig through the caption of the aggregated figure to discover which one corresponds to which insecticide.

We agree with the review comments. These result sections have been revised and the Figures updated providing headings to facilitate results interpretation (Figure 2, Lines 412-413; Figure 3, lines 498-499; Figure 4, Lines 559-560).

In the Discussion section, one of the key findings is that the frequency of the R allele is at high frequency or fixed in Central ("in Cameroon [...] the frequency of the 454A-CYP9K1 moved from 28.5% in 2014 to 94% in 2021") Africa while "no evidence of directional selection of this mutant-type 454A-CYP9K1 allele was observed in Southern Africa (Malawi and Mozambique)". This is simply incorrect. RR is almost fixed in one location in Cameroon (Mibellon) but at low to intermediate frequencies in the rest of the country indicating that the geographical distribution of the RR genotype might be more complex than that. Furthermore, most results show that the heterozygous genotype might confer enough resistance and the distribution in Southern Africa in 2020 or 2021 looks a lot like the distribution in Elende or Tibati in 2021. The same section also considers that it is possible that the resistance haplotype spread from Uganda to Cameroon. That seems possible but one would have to explain why this spread is only visible in 1 site in Cameroon and not the closest to Uganda. Thanks to the reviewer for this comment. We agree that the resistant allele is fixed in Mibellon and close to fixation in Penja while at moderate frequencies in other localities like Elende, as reported in Lines 605-608. In the temporal analysis section, we showed that despite the moderate frequency of the RR genotype in Elende, in 4 years the frequency of the mutant-type (R) allele significantly increased from 28.5% to 53% between 2019 and 2023 respectively (Lines 598-601). This was different situation in Malawi, where in 7 years the frequency of this mutant-type allele only changed by 1%, from 42% to 43% between 2014 and 2021 and in Mozambique from 34% to 33% between 2016 and 2020 (Lines 604-607). This is the justification of the directional selection of this mutant-type allele in Cameroon, but not in Southern Africa.

In some cases, the authors make their points in a confusing order. For instance, the bioassays are presented in the order alphacypermetrin, deltamethrin, permethrin; while the tube bio

assays are permethrin, alphacypermethrin (for the crosses); and alphacypermethrin, permethrin, deltamethrin (for the wild-caught).

We have corrected this to make the results more consistent and ensuring homogeneity in the presentation of results for the different assays (Lines 394-406, 456-473, 478-498).

Similarly, the discussion goes from "Allelic variation in CYP9K1 combines with its over-transcription to confer higher pyrethroid resistance intensity" (very much insecticide resistance focused) to "Novel G454A-CYP9K1 diagnostics are reliable DNA-based tools to detect and track the spread of metabolic resistance to pyrethroid insecticides" (about the importance of the assays) and back to "Mutant-type 454A-CYP9K1 allele reduces the efficacy of pyrethroid nets" (once again about insecticide resistance).

This discussion section has been corrected also to consistency and better flow of the ideas according to the reviewer comment. See discussion section (Lines 717-733).

Minor comments:

There are many situations where the English is either a little awkward (e.g., "predominant [...] than" l.623-624) or where the wrong word is used (e.g. "we unwound the role" l.861 is probably "we uncovered the role").

This section of the results has been corrected and rephrased accordingly (Lines 456-473 and Line 632). Thanks to reviewer for the remarks.

I think this sometimes stems from the authors avoiding to use the same phrase structure to make the manuscript easier to read but this leads to "Similarly" (l.744) being used to connect two sentences that literally show opposite behaviours and "Interestingly" (l.748) being used to observe that RS has an increased survivorship compared to SS, which has also been true for every other assay. The worst offender is probably "identified" at line 487 that can lead a reader to assume that this is the first time that anyone observed G454A-CYP9K1.

Thanks to the reviewer, corrections have been made. However, we will like to clarify that this is the first study establishing the role of G454A-CYP9K1 amino-acid change in conferring resistance in *An. funestus* and providing DNA-based diagnostic assay for field detection.

I have some issues with Figure 1 and the associated text:

In Figure 1A and B, only the dots representing the same country should be connected. As it is, the reader is left assuming that the relationship between, say Cameroon 2014 and Cameroon 2020 is similar to the one between Malawi 2020 and Ghana 2020.

I would agree that Figure 1A and B show a

"signature of strong directional selection" for Cameroon but it is less obvious for Uganda, where haplotype and nucleotide diversity were already quite low.

We have modified and updated this Figure 1 according to the reviewer comments (lines 360-361, Figure 1). Thanks to the reviewer for the remark.

The authors do not mention, in relation to Figure 1G, that 454A-

CYP9K1 is also present is also fixed for FUMOZ.

Thanks to the reviewer for this remark, we have corrected accordingly. But we would like to add that though the mutation is present in FUMOZ, the haplotypes are different (Figure S2a and b) from the dominant haplotype shared between Uganda and Cameroon (Figure 1f Lines 360-361).

The authors highlight the mutation G979A-CYP9K1 (exclusive to Ghana) for no clear reason.

Some other mutation (say A735-CYP9K1 or G738C-

CYP9K1) are fixed in Uganda, Cameroon and FUMOZ, present in Malawi but absent from FAN G and Ghana, a very similar pattern to G454A-CYP9K1. They should be mentioned

(in particular in light of the mention of G979A-

CYP9K1) if only to say that they are not the subject of this manuscript.

Many thanks to the reviewer for these remarks. Actually, we detected this SNP guanine (G) to adenine (A) nucleotide at position 979 (G979A-CYP9K1) causing change of amino acid valine to Isoleucine (I) on codon 279 exclusive to Ghana in 2020. We would like to add that we did not give priority to this V279I mutation here because previous studies (Riveron et al. 2017; Weedall et al. 2019) have shown that *CYP9K1* gene was not overexpressed in Western Africa, justifying why we started with the G454A-*CYP9K1* mutation detected in Eastern Africa where the gene has been shown to be overexpressed (Weedall et al. 2019). However, as we suggested, future studies could focus on validating the role of this V279I mutation (Line 787-788). Other mutations (A735-CYP9K1 or G738C-CYP9K1) as mentioned by the reviewer were not studied because they are silent mutations (not leading to amino acid changes) hence not affecting functional protein structure.

Figure 1E and F appear after G and H.

This has been modified and updated accordingly (lines 360-361, Figure 1) and Figure G and H transferred to the supplemental material (Figure S3a and b). Thanks to the reviewer.

I am not sure all readers will be very familiar with the various types of bednets and I think it would be helpful to remind, at the beginning of the section on EHTs and on the plots for the various bednets, which insecticide they use to better connect that section to the rest of the essays.

Thanks to the reviewer for this remark, we have corrected accordingly (lines 265-268, 281-282, 548, 554/55).

Times are sometimes shown as "hours" (l. 318, by the way, it should be "2 hours" and not "2hours"), "h" (l. 545) or "hrs" (l. 553). The authors should pick one and stick to it.

Thanks to the reviewer for highlighting this mistake which has been corrected accordingly. For example, in Lines 252, 397, 401, 480, 486, 494.

Some references (e.g., "WHO, 2022", l. 72) use a comma, others don't (e.g., "WHO 2022", l.76).

We have corrected this accordingly (Line 71).

A reference is needed l. 527.

A reference (Hearn *et al.* 2022) has been added (Line 385). Thanks to the reviewer for the remark.

The second sentence of "Mosquito samples collection and rearing" is barely readable. More than 1 sentence can be used and it might be worth pointing to the maps (Figures 5A and B) or create a new one.

Thanks for the remark, it has been corrected accordingly (Line 149 and Line 157).

For some reason, lines 329 and 330 are empty.

Thanks to the reviewer for pointing this out, it has been corrected accordingly (Line 228/229).

On line 601, I think "Figure 4A and B" is supposed to be "Figure 3A and B" as it is the reference at the end of the sentence. Furthermore, I don't think they are both necessary.

Correction done accordingly (Line 453).

The last 2 sentences of "Allele specific PCR (AS-PCR) design for G454A-CYP9K1 mutation" can probably be combined as they are very similar.

This has been corrected accordingly (Line 445-450), thanks to the reviewer for the remark.

On line 613, "In Contrary" is supposed to be "Contrarily", "However",
"On the other hand", or maybe "On the contrary".

All these have been corrected accordingly (Line 459).

On line 640, I assume that "454A-CYP9K1 mutant-type allele (S)" is supposed to be "454A-CYP9K1 mutant-type allele (R)".

This mistake has been corrected (Line 474).

On line 717, "the CYP9K1" is probably supposed to be "the CYP9K1 mutation" or "454A-CYP9K1".

This has been corrected accordingly (Line 541).

Article Title: A single mutation G454A in P450 CYP9k1 drives pyrethroid resistance in the major malaria vector *Anopheles funestus* reducing bed net efficacy

Personal Message to Authors:

I remain whole-heartedly impressed by the work in this paper and love the field-to-laboratory-to-field journey this endeavor takes us on as readers. On second review, I, unfortunately, cannot recommend the article for publication in its current form. The science is mostly ready, but the writing still is not in line with the quality of articles typically published by Genetics. I would like to commend you for your attempts to address the reviewer's comments from the first review. I could tell through those attempts that you showed a great deal of eagerness to get this manuscript ready for publication. It is my sincere hope that you will take the time to polish the writing and details in this article and resubmit the article to Genetics. Once ready, it will be a delight to our readers.

Reviewer Summary:

The authors take on an ambitious suite of aims in this study in an effort to first track the global regional distribution of a mutant allele – found in a vector fly population capable of carrying and spreading malaria – over time, then demonstrate the mutant allele's resistance to insecticide and test for a positive correlation or relationship between the expression of this allelic variant and resistance to widely used insecticides in Africa, design simple diagnostic tools to detect the resistance mutation in *An. funestus*, apply the diagnostics to track the efficacy of insecticide-treated bed nets in Africa, and, finally, offer an evaluation of what the presence, location, and fixation of this mutant allele means in the context of gene flow, microhabitats, and effective preventions for malaria across Africa. I have reduced all the components of the article into four main Aims: 1) Investigated genetic variation of the CYP9K1 gene in *An. funestus* across Africa and evaluated changes in allele frequencies over time, 2) through *in vitro* and *in vivo* experimental work, confirmed the relationship between mutant CYP9K1 and insecticide resistance and how that is impacted by gene expression, respectively, 3) designed diagnostic assays around the G454A mutation (mutant allele) and ran field studies on the efficacy of insecticide-treated bed nets, and 4) discussed what the increased resistance to insecticide means for *An. funestus* population genetics given gene flow between fly populations and connectivity of habitats. Rarely do we see a field-to-laboratory-to-field study contained in a single publication. As someone who appreciates full-circle, whole story science, I would like to commend the authors on bringing so many different elements of the story under the umbrella of one article.

I recognize all of your attempts to address the comments made in the first round of reviews and want to commend you for that. You also made a strong attempt to turn the paper around fast, and while I appreciate that, the nature of the first reviews really was to give you more time to organize and elevate the writing so that you could tell a cohesive and well-organized story for this manuscript that has a lot of moving parts. In the end, many of the adjustments did not serve to help that endeavor, and in some cases, like with the reorganization of the methods, weakened the article. The good news is that you still have a terrific paper, with terrific science, and once the writing is ready, you will absolutely find an audience here or elsewhere. I offer here some points

to help you bring the writing up to a level that aids your story and promotes the incredible amount of work you have done. I really do look forward to seeing this article in press.

Major Edits:

I had as many line item edits on the second review as I did not the first review and stopped recording them by the time I reached the results section. These included lines 65, 79, 80, 89-94, 94, 108, 110, 115, 128, 141, 149-157, 149 and 157, 161, 165, 167, 292/293, 303-308, etc. When an article has this many distractions, which might be as simple as a spelling issue, double reference, repeated word or line, or as complicated as a sentence that I struggle to understand, or is too long, or involves too many concepts, it detracts from where I need to be as a reader, which is the science. Given the enormous research undertaking here, this paper, even more so than most papers, has to be perfectly organized and easy for your reader to follow. There is just too much material to not do so, and you will lose your reader before they even make it out of methods. There are also some serious scientific writing rules that were violated in this version of the manuscript, including figure numbers that were out of order, an empty methods section (explained below), confusion about statistical tests associated with each experimental section (mentioned in first review), confusing population genetics language, and mis-use of biological and genetic terms or the confounding of terms due to extraneous language. There is an overall need for you to tighten the language of the article, make sentences clear, direct, and concise, remove unnecessary words, and provide your reviewers with clean, well-organized copy so that we can focus on the science.

On the first review, I had suggested two things to help the organization of the paper, including section headings for major experimental sections that were numbered to help the reader follow throughout the manuscript, and that you cut down the amount of words/pages in the methods section because it was extensive. I know you did that work, but, unfortunately, the decisions made in this case didn't aid the article. There were too many section headings. The use of main headings and sub-headings, in concert with the removal of methods information, meant that some sections had no more information than the headings themselves. I don't think the subheadings were necessary, but certainly you removed too much information from the Methods. Your reader still needs to be able to read your methods and be able to understand what the question/experiment was, how you performed it, and how you analyzed the results. This was still unclear and/or absent after the re-write in some cases.

Likewise, on first review, I had asked that for each experimental section that there be clear statistical tests reported in the methods for that section and clear statistical results reported in the same section in the Results. The changes here were also insufficient to aid your reader through the story of this research from field to laboratory to field.

Clear, concise language is needed throughout this paper. As one example, please revisit your lines 628-639. If this were re-written as the following, it would be more straightforward and effective: "Dissecting the genetic bases of insecticide resistance is imperative for control of malaria-transmitting mosquitoes and improving human health. We undertook a field-to-laboratory-to-field study to determine the role of P450 CYP9K1 gene in insecticide resistance in *An. funestus* mosquitoes and provide new diagnostics for tracking insecticide resistance in real-

time in this species.” This re-write is two short sentences. They are clear. They tell your reader what you did and why that was important. Doing this throughout the paper is a strong first step to the organization of this work and will provide your reader with something they can sail through and not feel like they are wading through.

I think the Introduction is ok, but it lacks a bit in organization and at times gets bogged down in details that may not be relevant for setting up the problem and providing brief background information. I think the Discussion is ok, but could be better organized and should include more information on how your results compare to alternative explanations, what they mean in a broader context of other genes promoting insecticide resistance, as well as differences between vector species, and definitely needs more thought about the studies limitations and future studies and applications of the knowledge gained by doing this work.

Doing this work now and before resubmitting here or elsewhere will only aid you, both in ease through a review process, but also in impact later once the manuscript is published, as I am confident that it will be published.

September 16, 2024

GENETICS-2024-307333

A single mutation G454A in P450 CYP9K1 drives pyrethroid resistance in the major malaria vector *Anopheles funestus* reducing bed net efficacy

Dear Dr. Djoko Tagne:

Upon second review, the reviewers still have comments and concerns that need to be addressed in a revised manuscript. It is most important that you address the quality of the written English in your manuscript. I believe the English editorial revisions needed for this manuscript are beyond what the editorial team at GENETICS will be able to assist on.

I appreciate that English may not be the primary language for the majority of your co-authors, but if any of them are native English speakers, this might be the easiest option -- please have one of them copy-edit to improve the quality of the English. If none of your co-authors are able to revise the quality of the English, you could explore professional services that would be able to assist you. You can find out more about these here.

https://academic.oup.com/pages/authoring/journals/preparing_your_manuscript/language_services

In the event that neither of these options is possible, please let me know. Finally, please also address the minor changes requested from Reviewer 3.

We look forward to receiving your revised manuscript.

Upon resubmission, please include:

1. A clean version of your manuscript;
2. A marked version of your manuscript in which you highlight significant revisions carried out in response to the major points raised by the editor/reviewers (track changes is acceptable if preferred);

Additionally, please ensure that your resubmission is formatted for GENETICS.

<https://academic.oup.com/genetics/pages/general-instructions>

Follow this link to submit the revised manuscript: Link Not Available

Sincerely,

Mara Lawniczak
Associate Editor
GENETICS

Approved by:
David Begun
Senior Editor
GENETICS

Reviewer #1 (Comments for the Authors (Required)):

Personal Message to Authors:

I remain whole-heartedly impressed by the work in this paper and love the field-to-laboratory-to-field journey this endeavor takes us on as readers. On second review, I, unfortunately, cannot recommend the article for publication in its current form. The science is mostly ready, but the writing still is not in line with the quality of articles typically published by Genetics. I would like to commend you for your attempts to address the reviewer's comments from the first review. I could tell through those attempts that you showed a great deal of eagerness to get this manuscript ready for publication. It is my sincere hope that you will take the time to polish the writing and details in this article and resubmit the article to Genetics. Once ready, it will be a delight to our readers.

Reviewer #3 (Comments for the Authors (Required)):

I am generally satisfied by the way the authors addressed my comments and I am thankful for their hard work.

I am, however, still not quite convinced that the evidence strongly supports their conclusions that "the fact that the same haplotype is predominant in both regions suggest that this 454A-CYP9K1 mutant allele (R) has spread from East to West across the Equatorial zone of the continent." and that "the high genetic polymorphism of the CYP9K1 gene observed in Malawi (Southern Africa) [...] highlight the little role played by the resistant allele of this gene in these regions." I would argue that the evidence strongly supports the first conclusion for part of Cameroon, however. 5 different sites in Cameroon provided samples that were used in this study:

- Mibellon is the one most consistent with the conclusion as it has seen a tremendous increase in frequency of the R allele and this haplotype is shown to be shared with Uganda and I thus grant that the conclusion is true for Mibellon.
- Penja has a very high frequency of the R allele in 2021 but no data is available for previous years. Ockham's razor would tend to conclude that the situation was similar in Penja to the one in neighbouring sites and I would thus agree that there has also been a great rise in frequency of the R allele there. That the haplotype is the same (and thus that it came from Uganda) is left unproved but I am willing to accept it as the simplest explanation.
- The situation in Elende is probably the most complex. If one only looks at the data for 2019 and 2021, the results are not miles of what is found in Malawi or Mozambique between 2014 and 2021 and 2016 and 2020, though admittedly on a shorter time scale. The results for 2023 are more striking and, I would agree, show an impressive increase in the frequency of the R allele but the fact that no data from South Africa is available for comparison isn't enough to conclude that the same process didn't occur there. Again the provenance is impossible to guess and the fact that the distribution of the R allele in Zambia in 2014 and in the DRC in 2021 looks similar to the one in Elende in 2023 cannot be ignored. Without haplotype data from any of these locations, it is impossible to be sure whether the haplotype in any of these sites is the same as the one from Uganda and thus whether Uganda is ultimately the origin of the mutation that is observed in Zambia, the DRC or southern Cameroon.
- The situation in Gounougou is very different. There is a clear increase in the frequency of the R allele but RR is still rare and R is still not as frequent as the S allele. There may be many explanation for it but it certainly doesn't support the conclusion as much as Mibellon did.
- Tibati has many failings of other sites: only one time point so it is impossible to estimate temporal trends; an underwhelming number of R alleles and a very small frequency RR; with the additional proximity to Mibellon making one wonder why the two sites are so different and question any conclusion that would cover a wide geographic region.

My recommendation would thus be to weaken a bit the conclusions by contrasting the results across Cameroon that give use different degrees of confidence in the spread of the R allele and by saying that the conclusion in South Africa may require more data to be confirmed.

Minor note:

The data availability statement is at the end of the manuscript instead of the end of the Methods but I don't think it is a problem.

Yaoundé, 14th October 2024

Dear Editor, *GENETICS* Journal

Ref: Research paper revision (Manuscript code Ref: GENETICS-2024-307333)

We are grateful to the editors/reviewers for their comments on the current manuscript titled "**A single mutation G454A in the P450 CYP9K1 drives pyrethroid resistance in the major malaria vector *Anopheles funestus* reducing bed net efficacy**", which greatly improved the quality of this revised manuscript. We have revised the entire manuscript according to the editors'/reviewers' comments and the list of changes made is presented below and also highlighted in the "track changes" version submitted.

Kindly find below the point-by-point responses to the comments.

Yours Sincerely,

Correspondence to:

Charles Sinclair Wondji
Liverpool School of Tropical Medicine (LSTM)
Pembroke Place, Liverpool L3 5QA, United Kingdom
Tel: +44 (0)151 705 3140/ +237 653 157 384
E-mail: charles.wondji@lstmed.ac.uk

Carlos Simeon Djoko Tagne
Centre for Research in Infectious Diseases (CRID)/ University of Bamenda
Nkoulou-Carriere, Yaounde, Cameroon
Tel: +237 651 060 395
E-mail: carlos.djoko@crid-cam.net

Reviewer #1 (Comments for the Authors (Required)):

Article Title: A single mutation G454A in P450 CYP9k1 drives pyrethroid resistance in the major malaria vector *Anopheles funestus* reducing bed net efficacy

Personal Message to Authors:

I remain whole-heartedly impressed by the work in this paper and love the field-to-laboratory-to field journey this endeavor takes us on as readers. On second review, I, unfortunately, cannot recommend the article for publication in its current form. The science is mostly ready, but the writing still is not in line with the quality of articles typically published by Genetics. I would like to commend you for your attempts to address the reviewer's comments from the first review. I could tell through those attempts that you showed a great deal of eagerness to get this manuscript ready for publication. It is my sincere hope that you will take the time to polish the writing and details in this article and resubmit the article to Genetics. Once ready, it will be a delight to our readers.

Authors appreciate this kind comment from Reviewer#1. Again, we appreciate the time and efforts the Reviewer invested for the feedback in this second review, which improved the quality of this manuscript.

Reviewer Summary:

The authors take on an ambitious suite of aims in this study in an effort to first track the global regional distribution of a mutant allele – found in a vector fly population capable of carrying and spreading malaria – over time, then demonstrate the mutant allele's resistance to insecticide and test for a positive correlation or relationship between the expression of this allelic variant and resistance to widely used insecticides in Africa, design simple diagnostic tools to detect the resistance mutation in *An. funestus*, apply the diagnostics to track the efficacy of insecticide treated bed nets in Africa, and, finally, offer an evaluation of what the presence, location, and fixation of this mutant allele means in the context of gene flow, microhabitats, and effective preventions for malaria across Africa. I have reduced all the components of the article into four main Aims: 1) Investigated genetic variation of the CYP9K1 gene in *An. funestus* across Africa and evaluated changes in allele frequencies over time, 2) through *in vitro* and *in vivo* experimental work, confirmed the relationship between mutant CYP9K1 and insecticide resistance and how that is impacted by gene expression, respectively, 3) designed diagnostic assays around the G454A mutation (mutant allele) and ran field studies on the efficacy of insecticide-treated bed nets, and 4) discussed what the increased resistance to insecticide means for *An. funestus* population genetics given gene flow between fly populations and connectivity of habitats. Rarely do we see a field-to-laboratory-to-field study contained in a single publication. As someone who appreciates full-circle, whole story science, I would like to commend the authors on bringing so many different elements of the story under the umbrella of one article. I recognize all of your attempts to address the comments made in the first round of reviews and want to commend you for that.

You also made a strong attempt to turn the paper around fast, and while I appreciate that, the nature of the first reviews really was to give you more time to organize and elevate the writing so that you could tell a cohesive and well-organized story for this manuscript that has a lot of moving parts. In the end, many of the adjustments did not serve to help that endeavor, and in some cases, like with the reorganization of the methods, weakened the article. The good news is that you still have a terrific paper, with terrific science, and once the writing is ready, you will absolutely find an audience here or elsewhere. I offer here some points to help you bring the writing up to a level that aids your story and promotes the incredible amount of work you have done. I really do look forward to seeing this article in press.

Major Edits:

I had as many lines item edits on the second review as I did not the first review and stopped recorded them by the time I reached the results section. These included lines 65, 79, 80, 89-94, 94, 108, 110, 115, 128, 141, 149-157, 149 and 157, 161, 165, 167, 292/293, 303-308, etc. When an article has this many distractions, which might be as simple as a spelling issue, double reference, repeated word or line, or as complicated as a sentence that I struggle to understand, or is too long, or involves too many concepts, it detracts from where I need to be as a reader, which is the science. Given the enormous research undertaking here, this paper, even more so than most papers, has to be perfectly organized and easy for your reader to follow. There is just too much material to not do so, and you will lose your reader before they even make it out of methods.

Thanks to the reviewer for pointing this out. The manuscript has been revised accordingly as can be seen in the track-change version, for example in Lines 70-74, 80-89, 95-110, 138-141, 143-145, 147-149, 151-153, 158-161, 166 for the Introduction, Lines 171-191, 196-197, 215-229, 232-246, 255-261, 266-291, 299-317, 321-376, 384-392, 406-410, 412-424 for the Materials and Methods, Lines 431-441, 481-487, 506-520, 539-578, 581-591, 658-674, 677-684, 726-750 for the results, 784-801, 806-834, 838-872, 874-900, 905-946, 949-975 for the Discussion 978-983, 985-992 and Lines 996-998 for the Conclusion and future Endeavours.

There are also some serious scientific writing rules that were violated in this version of the manuscript, including figure numbers that were out of order, an empty methods section (explained below), confusion about statistical tests associated with each experimental section (mentioned in first review), confusing population genetics language, and mis-use of biological and genetic terms or the confounding of terms due to extraneous language. There is an overall need for you to tighten the language of the article, make sentences clear, direct, and concise, remove unnecessary words, and provide your reviewers with clean, well-organized copy so that we can focus on the science.

We agree with the reviewer's comments and corrections have been made accordingly. The figures have been arranged in correct order, the language has been tightened and sentences made clear and the methodology has been revised and improved, for example Lines 40-50, 70-74, 80-89, 95-110, 138-141, 143-145, 147-149, 151-153, 158-161, 166 for the Introduction, Lines 171-191, 196-197, 215-229, 232-246, 255-261, 266-291, 299-317, 321-376, 384-392, 406-410, 412-424 for the Materials and Methods, Lines 431-441, 481-487, 506-520, 539-578,

581-591, 658-674, 677-684, 726-750 for the results, 784-801, 806-834, 838-872, 874-900, 905-946, 949-975 for the Discussion 978-983, 985-992 and Lines 996-998 for the Conclusion and future Endeavours.

On the first review, I had suggested two things to help the organization of the paper, including

section headings for major experimental sections that were numbered to help the reader follow throughout the manuscript, and that you cut down the amount of words/pages in the methods section because it was extensive. I know you did that work, but, unfortunately, the decisions made in this case didn't aid the article. There were too many section headings. The use of main headings and sub-headings, in concert with the removal of methods information, meant that some sections had no more information than the headings themselves. I don't think the subheadings were necessary, but certainly you removed too much information from the Methods. Your reader still needs to be able to read your methods and be able to understand what the question/experiment was, how you performed it, and how you analyzed the results. This was still unclear and/or absent after the re-write in some cases.

We highly appreciate these remarks from the reviewer. In the revised version, we have removed some sub-headings and revised the numbering by removing the sub-heading numberings and maintaining only the initial heading numberings, for example lines 171, 193, 209, 254, 271, 283, 292 and 384. Also, the methods section has been extensively revised, adding valuable information necessary for the reader to understand the experiments, for example lines 215-225, 232-246, 299-306 313-317 and 406-420,

Likewise, on first review, I had asked that for each experimental section that there be clear statistical tests reported in the methods for that section and clear statistical results reported in the same section in the Results. The changes here were also insufficient to aid your reader through the story of this research from field to laboratory to field.

Thanks to the reviewer for pointing this out and we have corrected this section accordingly to make statistical analysis section clear, Lines 405-421.

Clear, concise language is needed throughout this paper. As one example, please revisit your lines 628-639. If this were re-written as the following, it would be more straightforward and effective: "Dissecting the genetic bases of insecticide resistance is imperative for control of malaria-transmitting mosquitoes and improving human health. We undertook a field-to-laboratory-to-field study to determine the role of P450 CYP9K1 gene in insecticide resistance in *An. funestus* mosquitoes and provide new diagnostics for tracking insecticide resistance in realtime in this species." This re-write is two short sentences. They are clear. They tell your reader what you did and why that was important. Doing this throughout the paper is a strong first step to the organization of this work and will provide your reader with something they can sail through and not feel like they are wading through.

We appreciate these remarks from the reviewer. In the revised version, the English language throughout the manuscript has been made clear and concise for example, Lines 40-50, 80-89, 95-110, 157-161, for the Introduction, Lines 169-182, 255-259, 266-283, 287-293, 298-318, 323-334, 336-343, 347-361, 364-376, 384-392, 406-420, for the Methods, lines 430-443, 456-463, 481-487, 501-520, 537-587, 658-674, 725-750, for the Results, lines 784-800 806-821, 830-834, 848-872, 874-880, 912-926, 948-969, for the discussion and Lines 978-983, 985-992 for the Conclusion and future endeavours.

I think the Introduction is ok, but it lacks a bit in organization and at times gets bogged down in details that may not be relevant for setting up the problem and providing brief background information.

The introduction has been organised and some details removed in the revised version, for example Lines 68-75, 80-89, 95-110, 157-161.

I think the Discussion is ok, but could be better organized and should include more information on how your results compare to alternative explanations, what they mean in a broader context of other genes promoting insecticide resistance, as well as differences between vector species, and definitely needs more thought about the studies limitations and future studies and applications of the knowledge gained by doing this work.

We highly appreciate this remark and suggestion from the reviewer and we have reorganised the Discussion accordingly, for example Lines 784-800 806-821, 830-834, 848-872, 874-880, 912-926, 948-968.

Doing this work now and before resubmitting here or elsewhere will only aid you, both in ease through a review process, but also in impact later once the manuscript is published, as I am confident that it will be published.

Reviewer #3 (Comments for the Authors (Required)):

I am generally satisfied by the way the authors addressed my comments and I am thankful for their hard work.

I am, however, still not quite convinced that the evidence strongly supports their conclusions that

"the fact that the same haplotype is predominant in both regions suggest that this 454A-CYP9K1 mutant allele

(R) has spread from East to West across the Equatorial zone of the continent." and that

"the high genetic polymorphism of the CYP9K1 gene observed in Malawi (Southern Africa) [...] highlight the little role played by the resistant allele of this gene in these regions." I would argue that the evidence strongly supports the first conclusion for part of Cameroon, however, 5 different sites in Cameroon provided samples that were used in this study:

- Mibellon is the one most consistent with the conclusion as it has seen a tremendous increa

se in frequency of the R allele and this haplotype is shown to be shared with Uganda and I thus grant that the conclusion is true for Mibellon.

- Penja has a very high frequency of the R allele in 2021 but no data is available for previous years. Ockham's razor would tend to conclude that the situation was similar in Penja to the one in neighbouring sites and I would thus agree that there has also been a great rise in frequency of the R allele there. That the haplotype is the same (and thus that it came from Uganda) is left unproved but I am willing to accept it as the simplest explanation.

- The situation in Elende is probably the most complex. If one only looks at the data for 2019 and 2021, the results are not miles of what is found in Malawi or Mozambique between 2014 and 2021 and 2016 and 2020, though admittedly on a shorter time scale. The results for 2023 are more striking and, I would agree, show an impressive increase in the frequency of the R allele but the fact that no data from South Africa is available for comparison isn't enough to conclude that the same process didn't occur there. Again the provenance is impossible to guess and the fact that the distribution of the R allele in Zambia in 2014 and in the DRC in 2021 looks similar to the one in Elende in 2023 cannot be ignored. Without haplotype data from any of these locations, it is impossible to be sure whether the haplotype in any of these sites is the same as the one from Uganda and thus whether Uganda is ultimately the origin of the mutation that is observed in Zambia, the DRC or southern Cameroon.

- The situation in Gounougou is very different. There is a clear increase in the frequency of the R allele but RR is still rare and R is still not as frequent as the S allele. There may be many explanations for it but it certainly doesn't support the conclusion as much as Mibellon did.

- Tibati has many failings of other sites: only one time point so it is impossible to estimate temporal trends; an underwhelming number of R alleles and a very small frequency RR; with the additional proximity to Mibellon making one wonder why the two sites are so different and question any conclusion that would cover a wide geographic region.

My recommendation would thus be to weaken a bit the conclusions by contrasting the results across Cameroon that give use different degrees of confidence in the spread of the R allele and by saying that the conclusion in South Africa may require more data to be confirmed.

We highly appreciate this remark and suggestion from the reviewer #3 and corrections have been made accordingly, Lines 996-998.

Minor note:

The data availability statement is at the end of the manuscript instead of the end of the Methods but I don't think it is a problem.

We want to assure the reviewer that the data availability statement is at the end of the manuscript, Lines 999-1003.

October 24, 2024

RE: GENETICS-2024-307544

Dr. Carlos Simeon Djoko Tagne
Centre for Research in Infectious Diseases (CRID), Yaounde, Cameroon
Medical Entomology
Yaounde Cameroon
Yaounde, N/A +237
Cameroon

Dear Dr. Djoko Tagne:

Congratulations! We are delighted to inform you that your manuscript entitled "A single mutation G454A in the P450 CYP9K1 drives pyrethroid resistance in the major malaria vector *Anopheles funestus* reducing bed net efficacy" is acceptable for publication in GENETICS. Many thanks for submitting your research to the journal.

To Proceed to Production:

1. Format your article according to GENETICS style, as discussed at <https://academic.oup.com/genetics/pages/general-instructions>, and upload your final files at <https://genetics.msubmit.net>.
2. Your manuscript will be published as-is (unedited-as submitted, reviewed, and accepted) at the GENETICS website as an Advanced Access article and deposited into PubMed shortly after receipt of source files and the completed license to publish. Please notify sourcefiles@thegsajournals.org if you do not wish to publish your article via Advanced Access.
3. We invite you to submit an original color figure related to your paper for consideration as cover art. Please email your submission to the editorial office or upload it with your final files. You can submit a small-sized image for evaluation, and if selected, the final image must be a TIFF file 2513px wide by 3263px high (8.375 by 10.875 inches; resolution of 600ppi). Please avoid graphs and small type.

If you have any questions or encounter any problems while uploading your accepted manuscript files, please email the editorial office at sourcefiles@thegsajournals.org.

Sincerely,

Mara Lawniczak
Associate Editor
GENETICS

Approved by:
David Begun
Senior Editor
GENETICS

note: Please add jnls.author.support@oup.com and genetics.oup@kwglobal.com (or the domains @oup.com and @kwglobal.com) to your email program's "safe senders" list. You will be contacted by both at various points during the production process.